# A Plug-and-Play Query Synthesis Active Learning Framework for Neural PDE Solvers

**Zhiyuan Wang[1], Jinwoo Go[2], Byung-Jun Yoon[1,2], Nathan Urban[2], Xiaoning Qian[1,2,3]**

[1]Department of Electrical & Computer Engineering, Texas A&M University, College Station, TX
[2]Computing & Data Sciences, Brookhaven National Laboratory, Upton, NY
[3]Department of Computer Science & Engineering, Texas A&M University, College Station, TX

## Abstract

In recent developments in scientific machine learning (SciML), neural surrogate solvers for partial differential equations (PDEs) have become powerful tools for accelerating scientific computation for various science and engineering applications. However, training neural PDE solvers often demands a large amount of high-fidelity PDE simulation data, which are expensive to generate. Active learning (AL) offers a promising solution by adaptively selecting training data from the PDE settings–including parameters, initial and boundary conditions–that are expected to be most informative to help reduce this data burden. In this work, we introduce PaPQS, a **P**lug-**a**nd-**P**lay **Q**uery **S**ynthesis AL framework that synthesizes informative PDE settings directly in the continuous design space. PaPQS optimizes the Expected Information Gain (EIG) while encouraging batch diversity, enabling model-aware exploration of the design space via backpropagation through the neural PDE solution trajectories. The framework is applicable to general PDE systems and surrogate architectures, and can be seamlessly integrated with existing AL strategies. Extensive experiments across different PDE systems demonstrate that our AL framework, PaPQS, consistently improves sample efficiency over existing AL baselines.

## 1 Introduction

Partial differential equations (PDEs) are fundamental tools for modeling a wide range of physical phenomena in science and engineering. Traditional numerical solvers, such as finite difference and finite element methods [1], provide accurate solutions but often entail significant computational costs, especially when high spatial and temporal resolutions are required. The high computational cost becomes a critical bottleneck in applications involving fast or extensive evaluations, especially for consequent uncertainty quantification, optimization, and decision-making [2, 3].

Neural PDE solvers have recently been developed as efficient surrogates to alleviate this cost, enabling accelerated uncertainty analysis and decision making [4, 5, 6, 7]. Unlike typical supervised learning tasks, training neural PDE surrogates must account for spatiotemporal complexity and sensitivity to initial and boundary conditions, particularly in complex systems with chaotic or highly nonlinear dynamics. Therefore, randomly simulating many high-fidelity trajectories is impractical, given its prohibitive time and cost. These challenges motivate the development of new active learning (AL) methods tied to PDE systems, which seek to improve data efficiency by selectively querying the most informative PDE settings rather than relying on exhaustive data. Recent studies have evaluated AL for neural PDE surrogates, aiming to reduce the number of expensive simulations while maintaining neural surrogate model accuracy [8, 9, 10, 11, 12, 13].

39th Conference on Neural Information Processing Systems (NeurIPS 2025).

One representative class is adaptive active sampling approaches [10, 14, 15, 16], which prioritize training points based on PDE residuals estimated over continuous coordinates. However, these methods inherently rely on coordinate-based surrogates such as physics-informed neural network (PINN) [17, 18] and cannot be readily applied to widely used grid-based neural surrogates such as U-Net [7, 19]. Another class typically relies on pool-based selection strategies [8, 13, 20, 21, 22], which are constrained by the limited coverage of candidate pools and lack flexibility to explore the continuous design space. Moreover, since the pool is agnostic to the neural surrogates, these methods cannot dynamically respond to training progress or the model uncertainties, limiting the ability to refine queries. Furthermore, predictive variance is commonly used as the acquisition function in scientific modeling to evaluate data informativeness based on the model's epistemic uncertainty [8, 12, 21]. But maximizing the variance could lead to chaotic PDE solutions that distort acquisition signals, particularly under recurrent prediction settings [5, 7, 23]. Meanwhile, feature-based acquisition methods [13, 24, 25] emphasize input diversity but do not account for the relationship between PDE settings and model parameters. While recent pool-based methods have explored combining uncertainty- and diversity-based criteria [26, 27], they often require careful designs and remain limited to discrete candidate sets. There is a pressing need for a general mechanism that can flexibly plug into various acquisition strategies and enhance them by enabling continuous and adaptive query synthesis for the neural PDE solver problem.

To address these limitations, we propose a **P**lug-**a**nd-**P**lay **Q**uery **S**ynthesis (PaPQS) active learning framework for neural PDE solvers that directly synthesizes informative PDE settings in the continuous design space. Rather than depending on a fixed candidate pool, our framework optimizes a batch of candidate settings by ascending the expected information gain (EIG) – an acquisition function that directly interacts with the neural PDE solver and its parameters. By explicitly targeting the reduction of the posterior uncertainty, our framework dynamically explores high-informative regions while avoiding instability-prone areas that can mislead traditional uncertainty-based methods. We also introduce a policy function that incorporates an entropy-based regularization term to balance informativeness and batch diversity, encouraging wide coverage of the continuous search space and mitigating mode collapse. Importantly, PaPQS is designed in a plug-in style and can be readily combined with well-studied active learning strategies to further boost performance. This makes it a versatile framework that improves both the resolution and flexibility of sample acquisition, ultimately leading to more efficient training of neural PDE surrogates.

In summary, our main contributions include:

- We propose PaPQS that directly synthesizes informative PDE settings in the continuous search space, overcoming the rigidity of pool-based strategies. It considers both EIG and batch diversity, enabling adaptive exploration in the continuous space.

- We introduce an efficient critic-based approach to estimate EIG and its gradient with respect to the parametrized search space, allowing batch optimization to ascend informativeness. Our framework is extensible to general PDE systems and neural PDE surrogate architectures.

- Extensive experiments demonstrate that PaPQS generalizes well across diverse PDE systems and surrogate architectures, achieving consistent gains in sample efficiency and showing strong compatibility when combined with existing active learning strategies.

## 2 Background

### 2.1 Neural PDE surrogates

We focus on PDEs over a spatial domain $\mathcal{X}$ and temporal domain $[0, T]$, with solution $\mathbf{u}(t, \mathbf{x}) \in \mathbb{R}^{N_c}$, where $N_c$ denotes the number of physical variables or channels. Without loss of generality, the PDE formula can be written as follows:

$$\partial_t \mathbf{u} = F(\boldsymbol{\lambda}, t, \mathbf{x}, \mathbf{u}, \partial_x \mathbf{u}, \partial_{xx} \mathbf{u}, \dots), \quad (t, \mathbf{x}) \in [0, T] \times \mathcal{X} \tag{1}$$

$$\mathbf{u}(0, \mathbf{x}) = \mathbf{u}^0(\mathbf{x}), \quad \mathbf{x} \in \mathcal{X}; \quad \mathcal{B}[\mathbf{u}](t, \mathbf{x}) = 0, \quad (t, \mathbf{x}) \in [0, T] \times \partial\mathcal{X} \tag{2}$$

Here, $\mathcal{B}$ is the boundary condition. In this paper, we restrict our attention to periodic boundary conditions – the most commonly used setting in SciML studies [28] – for simplicity and in line with prior benchmarks [13]. $\boldsymbol{\lambda} \in \mathbb{R}^l$ denotes the PDE parameters (e.g., viscosity in the Navier–Stokes equations), and $\mathbf{u}^0$ represents the initial state of the system. We define the initial condition (IC),

$\boldsymbol{\psi} = (\mathbf{u}^0, \boldsymbol{\lambda})$, which includes both initial state and PDE parameters in this paper, could be drawn from a target input distribution $p_T(\boldsymbol{\psi}) = p_T(\mathbf{u}^0)p_T(\boldsymbol{\lambda})$. One popular choice for IC generator is that draws $\mathbf{u}^0$ from a superposition of sinusoidal functions with random parameters [28] and samples $\boldsymbol{\lambda}$ from a uniform distribution $\texttt{Uniform}(a, b)$. In this paper, we use this method to generate ICs and select or synthesize ICs via their generative parameters.

The ground-truth PDE solution is traditionally computed using a numerical solver that iteratively propagates the state forward in time, which is often computation-demanding and time-consuming. To accelerate such PDE simulations and enable real-time decision-making, neural surrogates have emerged as efficient alternatives [4, 7, 18, 29] and achieved success in various domains [8, 30, 31]. In this paper, we discuss the neural PDE solver with an autoregressive setting that predicts the next state as $\hat{\mathbf{u}}(t + \Delta t, \cdot) = \mathcal{M}_\theta(\hat{\mathbf{u}}(t, \cdot, \lambda))$. The training data consists of input-output pairs $(\boldsymbol{\psi}, \mathbf{u})$, where $\boldsymbol{\psi}$ is from the IC generator and $\mathbf{u}$ is the trajectory simulated by a numerical solver. The model is trained by minimizing the root mean squared error (RMSE):

$$\mathcal{L}_{\text{RMSE}}(\mathbf{u}, \hat{\mathbf{u}}) = \frac{1}{N_t N_x N_c} \sum_{i=1}^{N_t} \sum_{j=1}^{N_x} \|\mathbf{u}(t_i, x_j) - \hat{\mathbf{u}}(t_i, x_j)\|_2^2,$$

where $\hat{\mathbf{u}}$ is the predicted solution from the neural surrogate model, $N_t$ and $N_x$ denote the resolution of the discretized temporal and spatial domains, respectively. While this autoregressive setting serves as an example, we note that our proposed framework broadly applies to other types of neural PDE surrogates.

## 2.2 Expected information gain

The Expected Information Gain (EIG) is a principled acquisition criterion widely used in Bayesian optimal experimental design (BOED), where the goal is to select experimental conditions that maximize the information gained about unknown parameters [19]. With the IC input tuple $\boldsymbol{\psi}$, neural surrogate model parameters $\theta$, and observed solution fields $\mathbf{u}$, it is formally defined as:

$$EIG(\boldsymbol{\psi}) \triangleq \mathbb{E}_{p(\theta)p(\mathbf{u}|\theta, \boldsymbol{\psi})} \left[ \log \frac{p(\mathbf{u}|\theta, \boldsymbol{\psi})}{p(\mathbf{u}|\boldsymbol{\psi})} \right], \tag{3}$$

where $p(\theta)$ is the prior distribution over surrogate parameters $\theta$, $p(\mathbf{u}|\theta, \boldsymbol{\psi})$ is the likelihood function given a data point $\mathbf{u}$ at $\boldsymbol{\psi}$. This expression captures the expected reduction in posterior uncertainty and forms the basis of information-theoretic acquisition in BOED and active learning.

## 2.3 Active learning for neural PDE solvers

We now provide a formal problem definition of our active learning framework for neural PDE solvers. The goal is to train a more accurate neural PDE solver $\mathcal{M}_\theta$ using fewer labeled data points. Let $\mathcal{S}_{\text{train}}$ denote an initial training dataset, and $\mathcal{M}_\theta$ be the model trained on it. At each AL iteration, we aim to sample batches of informative ICs, $\mathcal{S}_{\text{itr}}$, query their labels (i.e., solve the numerical solvers), and augment the training set. This procedure is expected to yield faster convergence compared to randomly sampled ICs. We then retrain $\mathcal{M}_\theta$ on the updated dataset $\mathcal{S}_{\text{train}} \cup \mathcal{S}_{\text{itr}}$, and evaluate performance on a held-out test set with ICs drawn from the target distribution $p_T(\boldsymbol{\psi})$.

A standard practice in active learning for neural PDE solvers is the pool-based methods [8, 13, 21, 22, 32], where a candidate pool $\mathcal{S}_{\text{pool}}$ is randomly sampled prior to active learning. At each iteration, a subset of informative ICs $\mathcal{S}_{\text{itr}}$ is selected from $\mathcal{S}_{\text{pool}}$ according to a predefined acquisition function, labeled using numerical solvers, and added to the training set. The selected samples are subsequently removed from $\mathcal{S}_{\text{pool}}$. For the acquisition function considering interactions between model and data, the majority of existing studies rely on the predictive variance estimated either by model ensembles [13, 32] or inferred from the predictive distribution of probabilistic models [8].

## 3 Method

We now introduce our proposed AL framework PaPQS for neural PDE solvers, illustrated in Fig. 1. Instead of selecting from a fixed candidate pool, PaPQS operates directly in the continuous space of $\boldsymbol{\psi} = (\mathbf{u}^0, \boldsymbol{\lambda})$, synthesizing informative ICs by ascending the policy function. This design supports

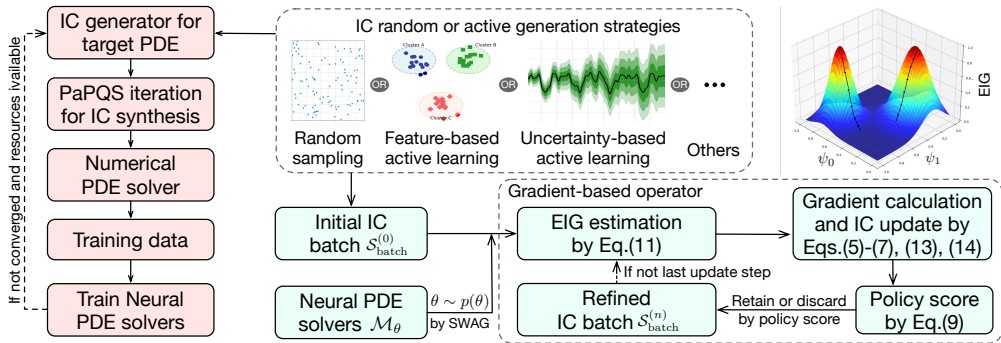

Figure 1: Overview of our PaPQS framework, including (a) the active learning pipeline for neural PDE solvers guided by PaPQS (red part) with (b) iterations of the PaPQS procedure (green part).

higher-resolution surrogate training, greater sampling flexibility, and more efficient data acquisition. Moreover, we aim for a plug-and-play design that can be readily applied to refine candidate ICs from existing AL strategies.

Based on this idea, we produce batches of ICs, $\mathcal{S}^*_{\text{batch}} = \{\boldsymbol{\psi}_1, \cdots, \boldsymbol{\psi}_{N_{\text{batch}}}\}$, with high informativeness – which is evaluated by the policy function – and label them for neural PDE surrogate's training at each AL iteration. Starting from an initial batch $\mathcal{S}^{(0)}_{\text{batch}}$, the IC synthesis process can be formulated as an optimization objective:

$$\mathcal{S}^*_{\text{batch}} = \arg\max_{\mathcal{S}_{\text{batch}}} \pi(\mathcal{S}_{\text{batch}}), \tag{4}$$

where $\pi(\cdot)$ is the policy function that quantifies the informativeness of a given batch through an acquisition measure and a diversity term. In our framework, this optimization is performed via a gradient-based update procedure with explicit control of batch diversity as described in Sec. 3.1. The acquisition measure in $\pi(\cdot)$ is defined as the EIG, whose estimation and gradient computation are detailed in Sec. 3.2.

## 3.1 Gradient-based operator

The optimization can be further designed to jointly capture individual informativeness and overall batch diversity. The synthesis of the next IC is guided towards regions of higher acquisition function scores, thereby improving the utility of individual new samples to acquire. Simultaneously, we impose a diversity-aware tradeoff to prevent the batch from collapsing into redundant regions, ensuring that the selected samples remain representative and diverse. This joint consideration enables us to synthesize training batches that are both informative and well-spread in the design space.

Specifically, we boost individual IC samples $\boldsymbol{\psi}$ via gradient ascent on the acquisition function using Adam optimizer [33], the single update step is

$$\boldsymbol{\psi}_i^{(n+1)} = \boldsymbol{\psi}_i^{(n)} + \alpha \cdot \frac{\hat{m}^{(n)}}{\sqrt{\hat{v}^{(n)}} + \epsilon}, \tag{5}$$

where $\mathcal{A}$ denotes the acquisition function that directly determines the gradient direction for ICs refinement, instantiated as the EIG in this work (see Sec. 3.2), $\hat{v}^{(n)} = \frac{\beta_2}{1-\beta_2} \cdot v^{(n-1)} + (\nabla_{\boldsymbol{\psi}_i^{(n)}} \mathcal{A}(\boldsymbol{\psi}_i^{(n)})^2$ and $\hat{m}^{(n)} = \frac{\beta_1}{1-\beta_1} \cdot m^{(n-1)} + \nabla_{\boldsymbol{\psi}_i^{(n)}} \mathcal{A}(\boldsymbol{\psi}_i^{(n)})$ are moment terms, $\alpha$ is the step size determining the magnitude of IC movement and $n \in [0 : N_{step}]$ denotes the $n$-th update step.

Eq. (5) facilitates an adaptive and gradient-based search procedure for active query synthesis in continuous design space, guiding each sample toward more informative regions while maintaining numerical stability through moment-based updates. In particular, a larger update step $N_{\text{step}}$ promotes broader exploration, allowing the optimizer to escape suboptimal basins and identify regions with higher expected information gain, while smaller step emphasize local exploitation, refining samples within high-EIG neighborhoods and maintaining batch-level diversity. This adaptive mechanism provides a principled trade-off between exploration and exploitation, crucial for efficient active learning on IC query synthesis.

To further enhance batch quality and ensure that the transformed samples are both informative and well-distributed, we introduce an entropy-based diversity regularization term, which is expressed by

$$\mathcal{H}(\boldsymbol{\psi}_i^{(n)}; \mathcal{S}_{\text{train}}, \mathcal{S}_{\text{batch}}^{(n)}) = \log 2D_k(\boldsymbol{\psi}_i^{(n)}; \mathcal{S}_{\text{train}}, \mathcal{S}_{\text{batch}}^{(n)}) - \frac{1}{d_{\boldsymbol{\psi}_i^{(n)}}} F(N_{D_k(\boldsymbol{\psi}_i^{(n)})} + 1). \tag{6}$$

Here, $\mathcal{H}(\cdot)$ could be regarded as an batch entropy estimator based on the k-nearest-neighbor and Kozachenko–Leonenko entropy [34, 35], with $D_k(\boldsymbol{\psi}_i^{(n)}; \mathcal{S}_{\text{train}}, \mathcal{S}_{\text{batch}})$ is the distance between $\boldsymbol{\psi}_i^{(n)}$ and its $k$-nearest neighbor in the space of $\mathcal{S}_{\text{train}} \cup \mathcal{S}_{\text{batch}}^{(n)}$, $N_{D_k(\boldsymbol{\psi}_i^{(n)})}$ denotes the number of training points whose distance to $\boldsymbol{\psi}_i^{(n)}$ is less than $D_k(\boldsymbol{\psi}_i^n)$, $d_{\boldsymbol{\psi}_i^{(n)}}$ is the dimensions of $\boldsymbol{\psi}_i^{(n)}$, and $F(\cdot)$ is the digamma function. This entropy term penalizes samples either densely clustered or located in regions already sufficiently represented by labeled data.

The policy function $\pi(\cdot)$ now consists of an acquisition term $\mathcal{A}(\cdot)$, which indicates informativeness, and a diversity regularization term $\mathcal{H}(\cdot)$, which measures spatial diversity. At each update step, it is calculated by

$$\pi(\boldsymbol{\psi}_i^{(n)}) = \mathcal{A}(\boldsymbol{\psi}_i^{(n)}) + \gamma \cdot \mathcal{H}(\boldsymbol{\psi}_i^{(n)}; \mathcal{S}_{\text{train}}, \mathcal{S}_{\text{batch}}^{(n)}), \tag{7}$$

where $\gamma$ is a hyperparameter balancing each term. Finally, we use this policy score to determine whether to retain each updated sample, i.e.,

$$\boldsymbol{\psi}_i^{(n+1)} = \begin{cases} \boldsymbol{\psi}_i^{(n+1)} & \text{if } \pi(\boldsymbol{\psi}_i^{(n+1)}) - \pi(\boldsymbol{\psi}_i^{(n)}) > \eta, \\ \boldsymbol{\psi}_i^{(n)} & \text{otherwise}, \end{cases} \tag{8}$$

where $\eta$ is a threshold that filters out non-contributive updates.

### 3.2 EIG-based acquisition function and gradient estimation

Previous works on active learning for neural PDE solvers often adopt predictive variance – typically estimated from ensembles [13, 21] or probabilistic models [8] – as the acquisition criterion. However, predictive variance can yield misleading acquisition signals in neural PDE settings for two reasons. First, many PDEs exhibit chaotic or highly sensitive behavior (e.g., the KS equation), where predictive variance may spike in response to small perturbations, without indicating true informativeness. Second, neural PDE solvers often adopt recurrent prediction schemes, where small prediction errors accumulate over time, further amplifying variance and distorting the acquisition signal [5, 7]. These effects can drive the synthesis toward uncertain yet unstable and uninformative regions of the IC space, ultimately degrading sample efficiency and model performance.

We adopt EIG (3) as our acquisition function as it effectively quantifies the expected uncertainty reduction over model parameters on potential observations, justified in Appendix A.1.

**Claim 1.** *Not all uncertainty is epistemically informative. It is therefore crucial to distinguish whether the observed uncertainty arises from a lack of knowledge about the model parameters (epistemic) or from intrinsic trajectory instability (aleatoric or chaotic) that cannot be reduced through learning [36]. Suppose predictive variance is used as the acquisition function. Then, preferred regions may reflect high prediction uncertainty (formulated as $\text{Var}_{\theta \sim p(\theta)}[\mathbb{E}[\mathbf{u} \mid \theta, \psi]]$), which can be dominated by chaotic sensitivity rather than epistemic uncertainty, and thus fail to provide useful information about $\theta$. In contrast, EIG explicitly quantifies the informativeness of $\psi$ with respect to updating the model parameters by minimizing posterior uncertainty, thereby prioritizing epistemically informative regions and supporting robust, data-efficient neural PDE surrogate learning*

Estimating EIG requires access to the likelihood function $p(\mathbf{u}|\theta, \boldsymbol{\psi})$ [37, 38]. However, since most neural PDE surrogates are implicit models without tractable likelihoods, we instead estimate the EIG by leveraging its equivalence to mutual information (MI). Specifically, we employ the MINE-f estimator as a lower bound on mutual information [39, 40]. This can be expressed as:

$$EIG(\boldsymbol{\psi}) \triangleq \text{MI}(\mathbf{u}|\boldsymbol{\psi}; \theta) = D_{\text{KL}}(p(\theta, \mathbf{u}) \| p(\theta)p(\mathbf{u}))$$
$$\geq \mathbb{E}_{p(\theta, \mathbf{u})}[T_\phi(\theta, \mathbf{u})] - e^{-1}\mathbb{E}_{p(\theta)p(\mathbf{u})}[e^{T_\phi(\theta, \mathbf{u})}], \tag{9}$$

where $T_\phi(\theta, \mathbf{u})$ is a neural critic function parameterized by $\phi$ with neural solver parameters $\theta$ and predictions $\mathbf{u}$ as input.

At each AL iteration, parameters $\phi$ could be optimized by maximizing (9) over the joint samples $(\theta_i, \mathbf{u}_i) \sim p(\theta)p(\mathbf{u}|\theta, \boldsymbol{\psi})$ and independent samples $\theta_i \sim p(\theta)$ and $\mathbf{u}_i \sim p(\mathbf{u}|\theta_j, \boldsymbol{\psi})$, where $\theta_j$ is another sample from $p(\theta)$ with $j \neq i$. To draw posterior parameter samples after training with the currently updated $p(\theta)$, an easy and popular way is to train an ensemble of surrogate models, where each model corresponds to a different approximate posterior sample [41]. Instead, we adopt a more efficient method, SWA-Gaussian (SWAG) [42], since it avoids the time-consuming ensemble training process, which is important for typical computationally constrained AL settings.

Using SWAG, the posterior samples $\theta_i$ are generated as follows:

$$\theta_i = \theta_{\text{SWA}} + \frac{1}{\sqrt{2}} \cdot \Sigma_{\text{diag}}^{\frac{1}{2}} z_{1,i} + \frac{1}{\sqrt{2(K-1)}} D z_{2,i}, \; z_{1,i} \sim \mathcal{N}(0, I_{d_\theta}), \; z_{2,i} \sim \mathcal{N}(0, I_{N_{\text{SWA}}}). \quad (10)$$

Here, we denote $\Theta = \{\theta_k\}_{k \in [1, N_{\text{SWA}}]}$ the trajectory of parameters $\theta$ in the last $N_{\text{SWA}}$ neural solver training epochs. Then $\theta_{\text{SWA}}$ is the mean value of $\Theta$. $\Sigma_{\text{diag}}$ is the variance calculated by $\Sigma_{\text{diag}} = \text{diag}(\frac{1}{N_{\text{SWA}}} \sum_{k=1}^{N_{\text{SWA}}} \theta_k^2 - \theta_{\text{SWA}}^2)$. $D$ is the deviation matrix comprised of columns $D_k = \theta_k - \theta_{\text{SWA}}$. For computational efficiency and to focus uncertainty estimation on the output space, we apply this sampling procedure solely to the output layer of the neural surrogate, keeping all preceding feature layers fixed, justified by the following proposition.

**Proposition 1.** *Let nerual PDE surrogate model parameters be partitioned as $\theta = (\theta_{feat}, \theta_{out})$, where $\theta_{out}$ is fixed and only $\theta_{out}$ is sampled. Assume that uncertainty over surrogate predictions $\mathbf{u}$ is dominated by the output layer parameters $\theta_{out}$, i.e., $p(\mathbf{u} \mid \theta, \boldsymbol{\psi}) = p(\mathbf{u} \mid \theta_{out}, \boldsymbol{\psi})$. Then, the expected information gain (EIG) can be approximated by sampling only $\theta_{out}$, i.e., $\text{EIG}(\psi) = I(\theta; \mathbf{u} \mid \boldsymbol{\psi}) = I(\theta_{out}; \mathbf{u} \mid \boldsymbol{\psi})$. A valid lower bound on the EIG can be estimated by sampling only $\theta_{out}$, given by:*

$$EIG(\boldsymbol{\psi}) \geq \mathbb{E}_{p(\theta_{out}, \mathbf{u})} \left[ T_\phi(\theta_{out}, \mathbf{u}) \right] - e^{-1} \mathbb{E}_{p(\theta_{out})p(\mathbf{u})} \left[ e^{T_\phi(\theta_{out}, \mathbf{u})} \right]. \quad (11)$$

Proof is provided in Appendix A.2. Posterior sampling can be performed without additional training cost based on (10), and can be computed in $\mathcal{O}(N_{\text{SWA}} d_\theta)$ time. We note that this approach could also be replaced by other efficient posterior approximation techniques, depending on practical considerations.

As for the sample of $\mathbf{u}$, it corresponds to the predictions derived by the neural PDE solver, given a sample pair $(\theta_i, \boldsymbol{\psi}_i)$. Here, we use the ICs $\boldsymbol{\psi} \in \mathcal{S}_{\text{batch}}^{(0)}$ as inputs to get predicted solutions $[\mathbf{u}(t, \boldsymbol{\psi}; \theta)]_{t \in [1:T], \boldsymbol{\psi} \in \mathcal{S}_{\text{batch}}^{(0)}}$ and form joint samples as $(\theta_i, \mathbf{u}(t, \boldsymbol{\psi}; \theta_i))$. Based on this setting, the training dataset for optimizing MINE-f is constructed to align the distribution of ICs to be updated, enabling it to effectively estimate the EIG during the gradient ascent process. To obtain the independent samples, we perform a random batch permutation over the set of joint samples.

With the MINE-f serving as a lower-bound estimator, we can now compute the gradient described in Eq. (5). Besides, we can also estimate the gradient with respect to the critic parameters $\phi$, i.e., $\nabla_\phi \mathcal{A}(\psi_i^{(n)})$. We therefore adopt an amortized update strategy [38] that jointly updates both the ICs $\boldsymbol{\psi}$ and the critic network $\phi$, which improves the accuracy and stability of EIG estimation throughout the gradient ascent process. The gradient at the $n$-th update step is then calculated as follows:

$$\nabla_{\boldsymbol{\psi}_i^{(n)}} \mathcal{A}(\boldsymbol{\psi}^{(n)}) = \nabla_{\boldsymbol{\psi}_i^{(n)}} \left( \mathbb{E}_{p(\theta, \mathbf{u})}[T_\phi(\theta, \mathbf{u})] - e^{-1} \mathbb{E}_{p(\theta)p(\mathbf{u})}[e^{T_\phi(\theta, \mathbf{u})}] \right)$$

$$\approx \frac{1}{N_t} \sum_{t=1}^{N_t} \nabla_{\boldsymbol{\psi}_i^{(n)}} \left( T_\phi(\theta_i, \mathbf{u}_i(t, \boldsymbol{\psi}_i; \theta_i)) - \frac{e^{-1}}{N_{\text{batch}}} \sum_{j=1}^{N_{\text{batch}}} e^{T_\phi(\theta_i, \mathbf{u}_j(t, \boldsymbol{\psi}_j; \theta_i))} \right)$$

$$= \frac{1}{N_t} \sum_{t=1}^{N_t} \nabla_{\mathbf{u}_i} T_\phi(\theta_i, \mathbf{u}_i(t, \boldsymbol{\psi}_i; \theta_i)) \cdot \nabla_{\boldsymbol{\psi}_i^{(n)}} \mathbf{u}_i(t, \boldsymbol{\psi}_i; \theta_i), \quad (12)$$

$$\nabla_\phi \mathcal{A}(\boldsymbol{\psi}^{(n)}) \approx \frac{1}{N_t} \sum_{t=1}^{N_t} \left( \nabla_\phi T_\phi(\theta_i, \mathbf{u}_i(t, \boldsymbol{\psi}_i; \theta_i)) - \frac{e^{-1}}{N_{\text{batch}}} \sum_{j=1}^{N_{\text{batch}}} \nabla_\phi e^{T_\phi(\theta_i, \mathbf{u}_j(t, \boldsymbol{\psi}_j; \theta_i))} \right). \quad (13)$$

For computational efficiency, we compute gradients only at three prediction steps – $\{1, \lfloor N_t/2 \rfloor, N_t\}$ – and average them to update the ICs. The complete procedure of our framework is provided in Appendix B.

# 4 Experiments

To discuss the efficacy and efficiency of the proposed PaPQS, we focus on the following research questions empirically: (1) **Performance**. Does PaPQS work? Fig. 2 compares PaPQS with state-of-the-art baselines using four commonlhy tested PDEs; (2) **Generalizability**. Does it support versatile neural PDE surrogate architectures? Fig. 3a verifies the adaptability of PaPQS across different models; (3) **Reusability.** Does PDE data synthesized by one architecture improve the training performance of other architectures? Fig. 3b presents a case under the cross-architecture setting; (4) **Efficiency.** Does it achieve comparable or superior performance with reduced training time? Figs. 3c and 3d present an 'accuracy vs. wall-clock time' analysis to demonstrate the data efficiency of the proposed PaPQS. (5) **Component.** How do components work? Figs. 4a and 4b conduct ablation studies to dissect the influence of each mechanism. (6) **Sensitivity.** What is the impact of varying parameters? Fig. 4c offers a sensitivity analysis of the key hyperparameters – the number of gradient update steps. Additional empirical results and case studies are provided in Appendix E.

## 4.1 Problem Setup

**PDEs, neural surrogates, and baselines.** We consider four PDEs with periodic boundary conditions as studied in a recent AL4PDE benchmark [13]: (1) **Burgers**' equation with viscosity [28]: $\partial_t \mathbf{u} + \mathbf{u}\partial_x \mathbf{u} = \left(\frac{\nu}{\pi}\right)\partial_{xx}\mathbf{u}$; (2) Kuramoto–Sivashinsky (**KS**) equation [11]: $\partial_t \mathbf{u} + \mathbf{u}\partial_x\mathbf{u} + \partial_{xx}\mathbf{u} + \nu\partial_{xxxx}\mathbf{u} = 0$; (3) Combined Equation (**CE**) [43] with the forcing term $\delta = 0$: $\partial_t \mathbf{u} + \partial_x \left(\alpha\mathbf{u}^2 - \beta\partial_x\mathbf{u} + \gamma\partial_{xx}\mathbf{u}\right) = 0$; (4) Compressible Navier-Stokes equation [28] in 2D space (**2D-CNS**): $\partial_t \rho + \nabla \cdot (\rho\mathbf{v}) = 0, \quad \rho\left(\partial_t\mathbf{v} + \mathbf{v}\cdot\nabla\mathbf{v}\right) = -\nabla p + \eta\Delta\mathbf{v} + (\zeta + \eta/3)\nabla(\nabla\cdot\mathbf{v})$, $\partial_t\left(\epsilon + \rho\mathbf{v}^2/2\right) + \nabla\cdot\left[\left(p + \epsilon + \rho\mathbf{v}^2/2\right)\mathbf{v} - \mathbf{v}\cdot\boldsymbol{\sigma}'\right] = 0$. We include three neural PDE surrogate solvers: (1) a modern U-Net [5] for PDE modeling; (2) SineNet [7] – an improved U-Net that mitigates feature misalignment by evolving high-resolution features through multiple sequential waves and achieves the state-of-the-art performance; (3) the Fourier Neural Operator [44] (FNO). The hyperparameters used for IC generation and neural PDE solvers are aligned with those in AL4PDE [13], with full details provided in Appendix C.

We compare our proposed AL framework with the following baselines: (1) **random** sampling and Latin Hypercube sampling (**LHS**) as static Design of Experiment (DoE) baselines; (2) pool-based AL strategies with predictive variance of an ensemble of models as the acquisition function. We select samples by **Top-K** and stochastic batch active learning [45] (**SBAL**) methods; (3) pool-based strategies with the feature-based input selection, including **CoreSet** [24] and **LCMD** [25]; (4) a hybrid strategy that first aggregates features using LCMD or Core-Set, and then applies **BAIT** [46] to minimize the average posterior predictive variance [25] over the aggregated features.

**Implementation details.** We adopt the same training settings for the neural PDE solvers as those used in the AL4PDE [13] benchmark. For the hyperparameters in the gradient-based operator (cf. Eqs. (5)-(8)), we perform ten gradient ascent steps for each batch of ICs, using a step size of $\alpha = 0.01$ scaled by the range of each parameter's range. We set the balance hyperparameter $\gamma$ to $0.5$, and the filter threshold $\eta$ to the median of the policy score computed in each batch. For EIG estimation, we record the parameter trajectory of the last 30 training epochs for the SWAG calculation in (10). As for the architecture of the neural critic function $T_\phi(\theta, \mathbf{u})$, we employ a convolutional module to reduce the dimension of input $\theta$ and $\mathbf{u}$, and then fuse them by a 3-layer MLP. Detailed training schedule and layer information are provided in Appendix D.

## 4.2 Performance comparison and model analysis

**Comparison of AL methods.** We report the average root-mean-square-error (RMSE) and 95% confidence interval in Fig. 2 and Tables E1- E4. Results show that our PaPQS consistently improves model performance over baseline AL strategies for neural PDE surrogates. Besides, combining PaPQS with existing AL methods (e.g., SBAL and LCMD) outperforms their stand-alone counterparts. For example for the Burgers' equations, our method reduces the average RMSE across iterations by 48.7%, 15.3%, and 7.4% when applied to candidate ICs obtained by random sampling, SBAL, and LCMD methods. This indicates that PaPQS provides the most benefit when paired with less informed strategies such as random sampling, with diminishing but consistent gains when combined with stronger baselines. This validates its strength as a general-purpose refinement module that enhances sample informativeness across diverse acquisition methods. The varying gains across

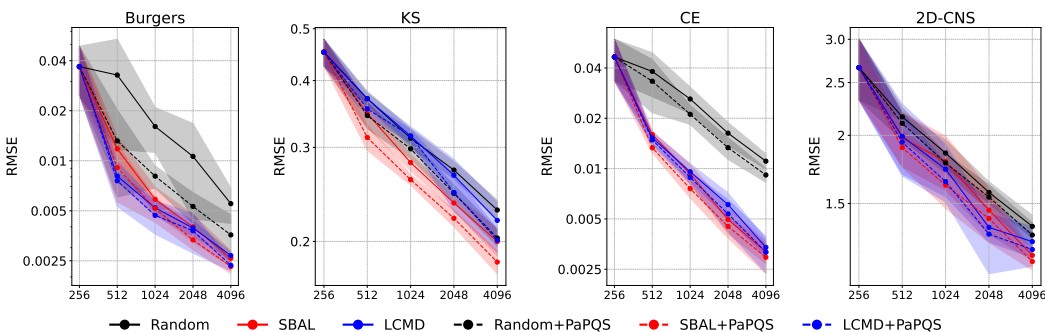

Figure 2: Error of U-Net over the number of trajectories in the training set. The shaded area represents the 95% confidence interval of the mean estimation calculated over multiple seeds.

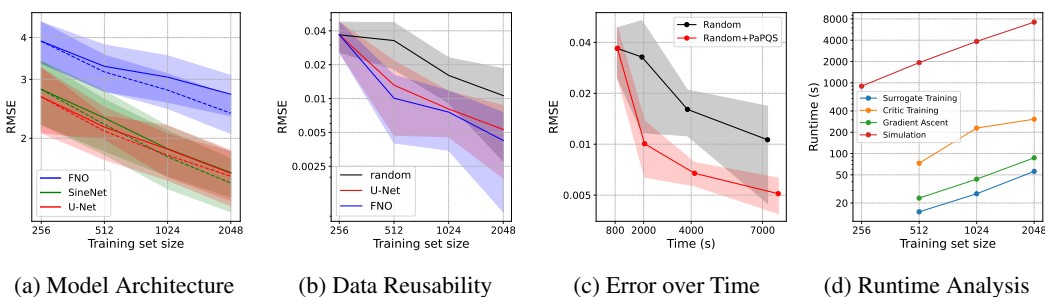

(a) Model Architecture     (b) Data Reusability     (c) Error over Time     (d) Runtime Analysis

Figure 3: (a) RMSE on 2D-CNS equation: solid and dashed lines denote the random and random+PaPQS strategies. (b) RMSE on Burgers' equation: U-Net is used as the evaluation model, with either U-Net or FNO as the acquisition model. (c) RMSE on Burgers' equation over time. (d) Runtime of each part of our PaPQS in the *"temporal behavior"* example.

PDEs can be attributed to differences in temporal and spatial complexity. For 2D-CNS equations, the improvement is moderate, which may be due to the system's complex, multi-variable coupling that makes informative IC optimization more challenging. Nevertheless, our method consistently provides stable performance improvement. With the results on Burgers' equations, characterized by simpler dynamics, showing pronounced benefits, the adaptability of our PaPQS framework has been demonstrated across diverse regimes including all these four PDE systems. We further evaluate our framework by changing the neural surrogate architecture. As shown in Fig.3a, PaPQS steadily reduces RMSE across all tested architectures, demonstrating its robustness and versatility.

**Data reusability.** We evaluate data reusability by testing whether a dataset synthesized using one neural surrogate architecture can benefit the training of another. Specifically, we train an FNO model to estimate the EIG and corresponding gradients, while retaining a U-Net as the evaluation model. Results are shown in Fig. 3b. Both U-Net and FNO-guided synthesis significantly outperform random sampling in downstream evaluation, even when the evaluation model (U-Net) differs from the model used for acquisition (FNO). This highlights the strong reusability and cross-model generalization of our IC synthesis, demonstrating the plug-and-play nature of our framework.

**Temporal behavior.** We evaluate the temporal behavior by tracking model performance over data generation time, which, for PaPQS, includes the total time consumption on training an acquisition model for EIG and related gradient estimations, training the neural critic function, gradient ascent, and simulation. Here, we estimate the acquisition function with an FNO with 30 training epochs, including 10 SWAG epochs for uncertainty estimation, and evaluate the performance with a U-Net model. Results in Fig. 3c indicate that PaPQS yields substantially lower RMSE than random sampling for the same data generation time budget. Although PaPQS introduces additional computations, these components account for less than 1/15 of the total runtime compared to simulation, as shown in Fig. 3d. The enhanced IC quality resulting from PaPQS leads to faster surrogate training convergence, underscoring its strong practical efficiency.

**Different acquisition functions.** We assess the impact of the acquisition function used in gradient ascent updates. While PaPQS employs EIG by default, we substitute it with predictive variance

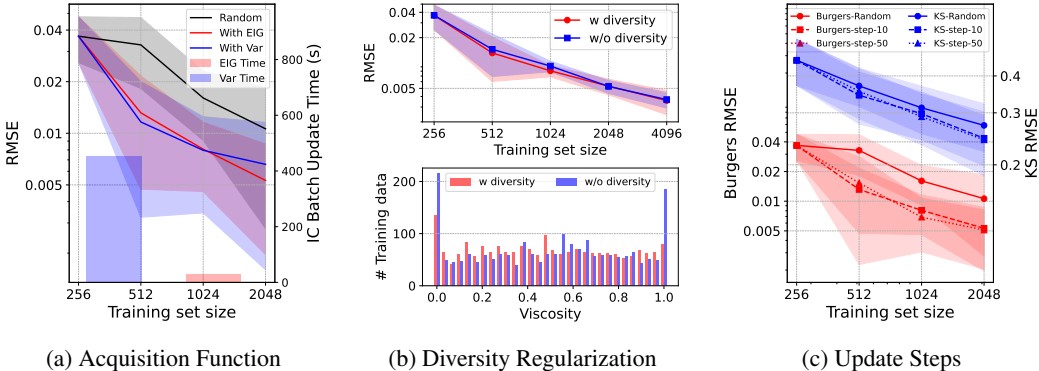

| (a) Acquisition Function | (b) Diversity Regularization | (c) Update Steps |

Figure 4: (a) RMSE and runtime for updating a batch of ICs on the Burgers' equation under different acquisition functions. (b) RMSE and queried PDE parameter (viscosity) distributions with and without diversity regularization on the Burgers' equation. (c) RMSE on the Burgers' and KS equations under different update step settings.

estimated from an ensemble of neural PDE solvers. The results are shown in Fig. 4a. Overall, both acquisition functions achieve comparable performance across all dataset sizes, consistently outperforming the random baseline. Notably, in some iterations, the predictive variance method exhibits slightly wider confidence intervals, suggesting it may introduce instability in AL. More importantly, our EIG method offers significantly lower computational cost. This stems from the gradient structure: EIG estimation in PaPQS relies on a contrastive formulation where gradients from negative samples can be safely truncated, since their trajectories are independent of the target IC (cf. Eq. (12)). In contrast, predictive variance requires multiple forward and backward passes for the same IC under different parameters, all of which contribute non-zero gradients. Consequently, optimizing variance incurs higher memory and runtime overhead, making EIG a more efficient choice for query synthesis.

**Effect of diversity regularization.** We evaluate the effectiveness of the proposed diversity regularization term in (6), with results presented in Fig. 4b. Incorporating the entropy-based term, PaPQS achieves 10% improvement in RMSE over its non-diversity counterpart during the first two iterations, while maintaining comparable performance in the latter two stages. This reflects the utility of the diversity term in scenarios with limited computational resources, where only a small number of labeled PDE simulations can be acquired and each query must be highly informative. As shown in the bottom panel of Fig. 4b, removing the diversity term causes the acquisition function to concentrate samples near the boundaries of the viscosity space – regions that typically exhibit high epistemic uncertainty but yield limited incremental information. This sampling bias results in poor coverage of the parameter space and hampers surrogate model training. In contrast, the proposed entropy-based regularization encourages broader and more balanced exploration across the viscosity domain, facilitating the acquisition of diverse dynamics and reducing redundancy in the training data. This leads to more sample-efficient learning and faster generalization with fewer labeled simulations.

**Effect of update steps.** We conduct a parameter sensitivity analysis to investigate how varying the number of gradient update steps $N_{step}$ affects the performance of our PaPQS framework. As shown in Fig. 4c, increasing the number of update steps from 10 to 50 yields minor improvements in RMSE on both the Burgers' and KS equations. This indicates that a small number of steps (e.g., $N_{step} = 10$) already achieves strong performance. Notably, the time complexity of the gradient operator scales linearly with the number of update steps. In practice, this parameter's setting can be flexibly adjusted based on computational resources and target performance, enabling a practical trade-off between efficiency and efficacy.

## 5 Conclusion

We present PaPQS, a plug-and-play active learning framework that synthesizes informative PDE settings through flexible exploration in continuous design space, different from existing pool-based AL methods for neural PDE solvers. By jointly optimizing expected information gain and batch diversity, PaPQS enables uncertainty-aware, adaptive query synthesis instead of relying on fixed candidate

pools, accelerating the training of neural PDE solvers. Experiments across diverse PDE systems and surrogate architectures show that PaPQS consistently improves data efficiency, generalizes across acquisition strategies, and achieves consistent gains in performance and runtime. While PaPQS performs well on simpler PDEs (e.g., Burgers' equation), its benefit is more modest on tightly coupled multi-field systems (e.g., 2D compressible Navier–Stokes). Future work may explore refined update strategies (e.g., adaptive step sizes or physics-informed regularization) to further improve such complex dynamics and extend active learning strategies to the temporal domain.

## 6 Acknowledgment

Z.W. and X.N.Q. acknowledge the support from United States National Science Foundation (NSF) grants DMREF-2119103, SHF-2215573, and IIS-2212419. J.G., B.J.Y., N.M.U., and X.N.Q. acknowledge the support by the U.S. Department of Energy's Office of Science Biological and Environmental Research (BER) program under project B&R# KP1601017 and FWP#CC140, and Advanced Scientific Computing Research (ASCR) under projects B&R# KJ0401010/FWP#CC130 and B&R# KJ0403010 and FWP#CC138.

Many of the numerical experiments were conducted using advanced computing resources provided by Texas A&M High Performance Research Computing.

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

# A  Theoretical Analysis

## A.1  Claim 1

*Suppose predictive variance is used as the acquisition function. Then, preferred regions may reflect high prediction uncertainty, suffering from strong chaos without necessarily providing information about θ. In contrast, EIG explicitly quantifies the informativeness of ψ with respect to updating the model parameters, thereby better supporting robust and data-efficient neural PDE surrogate learning.*

***Theoretical insight on linear case.*** To provide analytical intuition, we consider a simplified linear surrogate model

$$\mathbf{u}_{t+1} = \boldsymbol{\psi}w + \epsilon,$$

where $\boldsymbol{\psi}$ denotes the design variable (e.g., initial condition), $w \sim \mathcal{N}(0, \Sigma_w)$ represents the model parameters, and $\epsilon \sim \mathcal{N}(0, \sigma^2)$ is the additive noise capturing aleatoric or measurement uncertainty.

The expected information gain is defined as the mutual information between $\mathbf{u}$ and $w$ given $\boldsymbol{\psi}$:

$$EIG(\boldsymbol{\psi}) = I(w; \mathbf{u} \mid \boldsymbol{\psi}) = H(\mathbf{u} \mid \boldsymbol{\psi}) - H(\mathbf{u} \mid w, \boldsymbol{\psi}).$$

Under the linear model, we have

$$\mathbf{u} \mid w, \boldsymbol{\psi} \sim \mathcal{N}(\boldsymbol{\psi}w, \sigma^2), \quad w \sim \mathcal{N}(0, \Sigma_w),$$

which implies the marginal distribution

$$\mathbf{u} \mid \boldsymbol{\psi} \sim \mathcal{N}(0, \boldsymbol{\psi}\Sigma_w\boldsymbol{\psi}^\top + \sigma^2).$$

Hence, the conditional entropies can be computed as

$$H(\mathbf{u} \mid w, \boldsymbol{\psi}) = \tfrac{1}{2}\log(2\pi e\sigma^2), \quad H(\mathbf{u} \mid \boldsymbol{\psi}) = \tfrac{1}{2}\log(2\pi e(\boldsymbol{\psi}\Sigma_w\boldsymbol{\psi}^\top + \sigma^2)).$$

Subtracting the two gives

$$\begin{aligned}
EIG(\boldsymbol{\psi}) &= H(\mathbf{u} \mid \boldsymbol{\psi}) - H(\mathbf{u} \mid w, \boldsymbol{\psi}) \\
&= \tfrac{1}{2}\log(2\pi e(\boldsymbol{\psi}\Sigma_w\boldsymbol{\psi}^\top + \sigma^2)) - \tfrac{1}{2}\log(2\pi e\sigma^2) \\
&= \tfrac{1}{2}\log\left(\frac{\boldsymbol{\psi}\Sigma_w\boldsymbol{\psi}^\top + \sigma^2}{\sigma^2}\right) \\
&= \tfrac{1}{2}\log\left(1 + \frac{\boldsymbol{\psi}\Sigma_w\boldsymbol{\psi}^\top}{\sigma^2}\right).
\end{aligned}$$

This closed form shows that $EIG(\boldsymbol{\psi})$ depends purely on the epistemic uncertainty term $\boldsymbol{\psi}\Sigma_w\boldsymbol{\psi}^\top$, while the additive noise variance $\sigma^2$ only acts as a normalization constant in the logarithmic term.

Similarly, the predictive variance can be expressed as

$$V[\mathbf{u} \mid \boldsymbol{\psi}] = V_w[\boldsymbol{\psi}w] + V_\epsilon[\epsilon] = \boldsymbol{\psi}\Sigma_w\boldsymbol{\psi}^\top + \sigma^2.$$

Unlike $EIG(\boldsymbol{\psi})$, the predictive variance includes both epistemic and aleatoric contributions, and can therefore be inflated by stochastic noise unrelated to model learnability. This distinction clarifies why EIG serves as a more principled acquisition objective for active learning—it explicitly measures the expected reduction in parameter uncertainty. In contrast, predictive variance may overemphasize chaotic or noisy regions.

As for the optimization property, since $\Sigma_w \succ 0$ ensures that the quadratic form $\boldsymbol{\psi}\Sigma_w\boldsymbol{\psi}^\top$ is convex and the outer function $\log(1+x)$ is strictly concave and monotonic, the composition $EIG(\boldsymbol{\psi})$ defines a concave objective. Thus, maximizing $EIG(\boldsymbol{\psi})$ corresponds to a convex optimization problem (maximization of a concave function), for which gradient ascent guarantees convergence to the global optimum in this linear regime.

Although the exact closed-form EIG is intractable in nonlinear PDE surrogates, this linear analysis provides theoretical support for the gradient-based optimization strategy adopted in PaPQS. It also explains why the EIG-based critic naturally filters out uninformative, noise-dominated regions and promotes robust and data-efficient query synthesis in complex PDE settings.

***Illustrative example.*** To support our claim that the expected information gain (EIG) is more reliable than predictive variance in selecting informative samples under chaotic PDE dynamics, we present both visual and quantitative evidence based on an active learning case study of the Kuramoto–Sivashinsky (KS) equation. The detailed settings are as follows:

- PDE: we consider the 1D KS equation, a canonical model exhibiting spatiotemporal chaotic dynamics, defined as $\frac{\partial \mathbf{u}}{\partial t} + \mathbf{u}\frac{\partial \mathbf{u}}{\partial x} + \frac{\partial^2 \mathbf{u}}{\partial x^2} + \nu \frac{\partial^4 \mathbf{u}}{\partial x^4} = 0$. The equation is solved under periodic boundary conditions on a spatial domain of length $L = 64$, discretized using $N_x = 256$ spatial grid points. Time integration is performed over $N_t = 800$ steps with a fixed time step size $\Delta t = 0.01$;

- Initial Condition (IC) $\boldsymbol{\psi}$: the viscosity parameter $\nu$ is sampled from the range $[0.05, 0.1]$. $\mathbf{u}^0$ is constructed as a sum of random sinusoidal modes to induce variability and richness in dynamical behavior, which could be formulated as $\mathbf{u}^0 = \sum_{i=1}^{N_w} A_i \sin\left(2\pi k_i x/L + \phi_i\right)$. Here, $N_w$ is the wavenumber, $A_i$ is the amplitude ranging from $[-1, 1]$, $k_i$ is the frequency sampled from $[1, 10]$, and $\phi_i$ is the phase from $[0, 2\pi]$. We perform uniform sampling for all parameters to determine initial conditions;

- Surrogate model $\mathcal{M}_\theta$: We adopt a probabilistic autoregressive neural network as the surrogate model to learn the evolution dynamics of the KS equation. Given the state $u_t$ at time step $t$, the model predicts the distribution of the next state $\mathbf{u}_{t+1}$ by outputting the mean and log-variance of a Gaussian distribution, i.e., $p_\theta(\mathbf{u}_{t+1}|\mathbf{u}_t) = \mathcal{N}(\mu_\theta(\mathbf{u}_t), \mathrm{diag}(\sigma_\theta^2(\mathbf{u}_t)))$, where both $\mu_\theta(\cdot)$ and $\log \sigma_\theta^2(\cdot)$ are parameterized by a shared multilayer perceptron (MLP) with two hidden layers and dropout regularization. The model is trained with a Gaussian negative log-likelihood (NLL) loss:

$$\mathcal{L}_{\mathrm{NLL}} = \frac{1}{2} \sum_{i=1}^{N_x} \sum_{t=1}^{N_t} \left[ \log \boldsymbol{\sigma}^2(t+1, i) + \frac{(\mathbf{u}(t+1, i) - \boldsymbol{\mu}(t+1, i))^2}{\boldsymbol{\sigma}^2(t+1, i)} \right]. \qquad (14)$$

To reduce computational cost while preserving temporal dynamics, we train the model to predict an *up-sampled* trajectory by selecting every 40 frames from the original simulation, resulting in 20 autoregressive prediction steps.

To directly compare the difference between two acquisition functions, we consider a pool-based active learning scheme, which follows the steps below:

1. We randomly sample 30 trajectories from the PDE simulator to form an initial labeled set. 10 trajectories are used to train the surrogate model, and another 20 trajectories are held as the test set;

2. Next, we construct a candidate pool of 200 unlabeled samples by varying the ICs within the prescribed design space;

3. Each candidate is scored using one of the two acquisition functions – predictive variance or EIG – and the top 10 samples are selected and added to the training set. Their estimation methods are provided in Algorithms A1 and A2;

4. The surrogate model is then fine-tuned on the augmented dataset, and its performance is evaluated on the test set.

This single-round selection process enables a direct and fair comparison of the informativeness of the two acquisition strategies under identical conditions.

We repeat this experiment 10 times with different random seeds and report the RMSE and 95% confidence interval after training on the selected PDE data. The variance-based method yields an RMSE of **4.748±0.116**, while the EIG-based selection achieves a lower RMSE of **4.688±0.092**. The EIG-guided AL achieves lower errors, indicating more effective sample selection for model improvement. We further visualize ground-truth PDE trajectories selected by predictive variance and EIG-based acquisition functions in Fig. A1 to investigate the types of dynamics favored by each method. We observe that predictive variance tends to select highly chaotic trajectories characterized by rapidly changing modes, spatial discontinuities, and temporal irregularities (e.g., samples 19, 141, and 42). This is expected, as predictive variance reacts strongly to output variability, which is amplified in chaotic regions. However, such samples – despite their high uncertainty – may be

---

**Algorithm A1** Predictive Variance Estimation via Dropout Sampling

---

**Input:** Initial condition $\psi$, surrogate model $\mathcal{M}_\theta$ with dropout, number of samples $N_s$, prediction horizon $N_t$

**Output:** Predictive variance estimate $\boldsymbol{\sigma}^2_{\text{total}}$

1: **for** $s \in [1, N_s]$ **do**
2:      Randomly sample a model $\mathcal{M}_\theta^{(s)}$ by Dropout
3:      Get mean and variance predictions $\{\boldsymbol{\mu}^{(s)}(t, x), \boldsymbol{\sigma}^{(s)^2}(t, x)\}_{t \in [1, N_t]}$ via autoregressive rollout
4: **end for**
5: Get epistemic variance by computing the empirical variance of mean predictions, i.e., $\boldsymbol{\sigma}^2_{\text{epistemic}} \leftarrow \text{Var}_s \left[ \{\boldsymbol{\mu}^{(s)}\} \right]_{s \in [1, N_s]}$
6: Get aleatoric variance by averaging the variance prediction, i.e., $\boldsymbol{\sigma}^2_{\text{aleatoric}} \leftarrow \mathbb{E}_s[\{\boldsymbol{\sigma}^{(s)^2}\}_{s \in [1, N_s]}]$
7: Get total variance by adding the epistemic and aleatoric variance: $\boldsymbol{\sigma}^2_{\text{total}} \leftarrow \boldsymbol{\sigma}^2_{\text{epistemic}} + \boldsymbol{\sigma}^2_{\text{aleatoric}}$
8: **return** $\boldsymbol{\sigma}^2_{\text{total}}$

---

**Algorithm A2** Expected Information Gain Estimation via Nested Monte Carlo

---

**Input:** Initial condition $\psi$, surrogate model $\mathcal{M}_\theta$ with dropout, number of outer samples $N_s$, number of inner samples $N_m$, prediction horizon $N_t$

**Output:** Estimated expected information gain $EIG(\psi)$

1: Initialize total EIG estimate $EIG \leftarrow 0$
2: **for** $s \in [1, N_s]$ **do**
3:      Sample a model $\mathcal{M}_\theta^{(s,0)}$ by Dropout
4:      Get mean and variance predictions $\{\boldsymbol{\mu}^{(s,0)}(t, x), \boldsymbol{\sigma}^{(s,0)^2}(t, x)\}_{t \in [1, N_t]}$ via autoregressive rollout
5:      Sample a trajectory $\mathbf{u}^{(s)}$ from $\mathcal{N}(\boldsymbol{\mu}^{(s,0)}, \boldsymbol{\sigma}^{(s,0)^2})$
6:      Compute log-likelihood via $\log p(\mathbf{u}^{(s)}|\theta_{s,0}) \leftarrow \log \mathcal{N}(\mathbf{u}^{(s)}; \boldsymbol{\mu}^{(s,0)}, \boldsymbol{\sigma}^{(s,0)^2})$
7:      **for** $m \in [1, N_m]$ **do**
8:          Sample another model $\mathcal{M}_\theta^{(s,m)}$ by Dropout
9:          Get mean and variance predictions $\{\boldsymbol{\mu}^{(s,m)}(t, x), \boldsymbol{\sigma}^{(s,m)^2}(t, x)\}_{t \in [1, N_t]}$ via autoregressive rollout
10:         Compute log-likelihood via $\log p(\mathbf{u}^{(s)}|\theta_{s,m}) \leftarrow \log \mathcal{N}(\mathbf{u}^{(s)}; \boldsymbol{\mu}^{(s,m)}, \boldsymbol{\sigma}^{(s,m)^2})$
11:      **end for**
12:      Estimate marginal log-likelihood by $\log p(\mathbf{u}^{(s)}) \leftarrow \frac{1}{M} \sum_{m=1}^M p(\mathbf{u}^{(s)} \mid \theta_{s,m})$
13: **end for**
14: Estimate the EIG by $EIG(\psi) \leftarrow \frac{1}{N_s} \sum_{s=1}^{N_s} \frac{\log p(\mathbf{u}^{(s)}|\theta_{s,0})}{\log p(\mathbf{u}^{(s)})}$
15: **return** $EIG(\psi)$

---

less informative for model training due to their intrinsic unpredictability. In contrast, EIG tends to prioritize relatively stable trajectories that are still dynamically rich but exhibit more learnable patterns, avoiding excessively chaotic regions that may hinder learning. This allows the surrogate model to extract more meaningful gradients, leading to faster convergence under a limited sample budget.

## A.2 Proposition 1

*Let neural PDE surrogate model parameters be partitioned as $\theta = (\theta_{feat}, \theta_{out})$, where $\theta_{out}$ is fixed and only $\theta_{out}$ is sampled. Assume that the dominant sources of epistemic uncertainty of the model can be captured by $\theta_{out}$ [47]. Then, a valid lower bound on the EIG can be estimated by sampling only $\theta_{out}$, given by: $EIG(\psi) \geq \mathbb{E}_{p(\theta_{out}, \mathbf{u})} \left[ T_\phi(\theta_{out}, \mathbf{u}) \right] - e^{-1} \mathbb{E}_{p(\theta_{out})p(\mathbf{u})} \left[ e^{T_\phi(\theta_{out}, \mathbf{u})} \right]$,*

*Proof.* We start from the definition of expected information gain (EIG) under the full model parameter space $\theta = (\theta_{\text{feat}}, \theta_{\text{out}})$. Given input condition $\psi$ and model prediction $\mathbf{u}$, EIG is defined as:

$$\text{EIG}(\psi) = \mathbb{E}_{\theta \sim p(\theta)} \left[ \text{KL} \left( p(\mathbf{u} \mid \theta, \psi) \,\|\, p(\mathbf{u} \mid \psi) \right) \right],$$

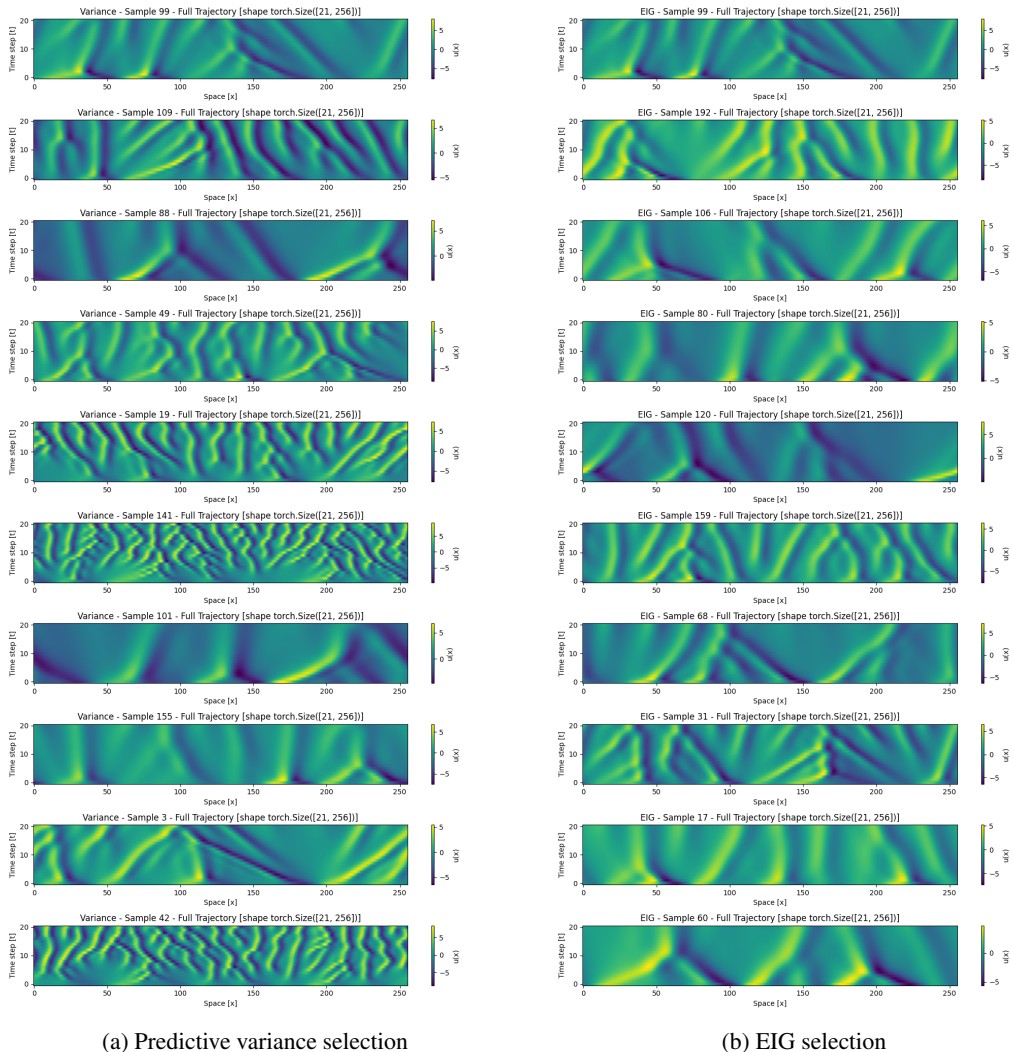

(a) Predictive variance selection          (b) EIG selection

Figure A1: The visualization of ground-truth PDE trajectories selected by different acquisition functions.

where $p(\mathbf{u} \mid \boldsymbol{\psi}) = \int p(\theta)p(\mathbf{u} \mid \theta, \boldsymbol{\psi})d\theta$ is the marginal predictive distribution.

Now, we assume that the feature extractor parameters $\theta_{\text{feat}}$ are fixed to some constant $\theta_{\text{feat}}^*$. This implies that the prior over parameters becomes:

$$p(\theta_{\text{feat}}, \theta_{\text{out}}) = \delta(\theta_{\text{feat}} - \theta_{\text{feat}}^*) \cdot p(\theta_{\text{out}}), \tag{15}$$

and the predictive distribution simplifies accordingly:

$$p(\mathbf{u} \mid \theta, \boldsymbol{\psi}) = p(\mathbf{u} \mid \theta_{\text{feat}}^*, \theta_{\text{out}}, \boldsymbol{\psi}) := p(\mathbf{u} \mid \theta_{\text{out}}, \boldsymbol{\psi}). \tag{16}$$

Substituting this into the definition of EIG, we obtain:

$$
\begin{aligned}
\mathrm{EIG}(\boldsymbol{\psi}) &= \mathbb{E}_{(\theta_{\mathrm{feat}}, \theta_{\mathrm{out}}) \sim p(\theta)} \left[ \mathrm{KL} \left( p(\mathbf{u} \mid \theta_{\mathrm{feat}}, \theta_{\mathrm{out}}, \boldsymbol{\psi}) \,\|\, p(\mathbf{u} \mid \boldsymbol{\psi}) \right) \right] \\
&= \int \int \delta(\theta_{\mathrm{feat}} - \theta_{\mathrm{feat}}^*) \cdot p(\theta_{\mathrm{out}}) \cdot \mathrm{KL} \left( p(\mathbf{u} \mid \theta_{\mathrm{feat}}, \theta_{\mathrm{out}}, \boldsymbol{\psi}) \,\|\, p(\mathbf{u} \mid \boldsymbol{\psi}) \right) \, d\theta_{\mathrm{out}} \, d\theta_{\mathrm{feat}} \\
&= \int p(\theta_{\mathrm{out}}) \cdot \mathrm{KL} \left( p(\mathbf{u} \mid \theta_{\mathrm{feat}}^*, \theta_{\mathrm{out}}, \boldsymbol{\psi}) \,\|\, p(\mathbf{u} \mid \boldsymbol{\psi}) \right) \, d\theta_{\mathrm{out}} \\
&= \int p(\theta_{\mathrm{out}}) \cdot \mathrm{KL} \left( p(\mathbf{u} \mid \theta_{\mathrm{out}}, \boldsymbol{\psi}) \,\|\, p(\mathbf{u} \mid \boldsymbol{\psi}) \right) \, d\theta_{\mathrm{out}} \\
&= \mathbb{E}_{\theta_{\mathrm{out}} \sim p(\theta_{\mathrm{out}})} \left[ \mathrm{KL} \left( p(\mathbf{u} \mid \theta_{\mathrm{out}}, \boldsymbol{\psi}) \,\|\, p(\mathbf{u} \mid \boldsymbol{\psi}) \right) \right] \\
&= \texttt{Mutual-Information}(\theta_{\mathrm{out}}, \mathbf{u} \mid \boldsymbol{\psi}) \\
&\geq \mathbb{E}_{p(\theta_{\mathrm{out}}, \mathbf{u})} [T_\phi(\theta_{\mathrm{out}}, \mathbf{u})] - e^{-1} \mathbb{E}_{p(\theta_{\mathrm{out}}) p(\mathbf{u})} [e^{T_\phi(\theta_{\mathrm{out}}, \mathbf{u})}].
\end{aligned}
\tag{17}
$$

This completes the proof.

$\square$

## B  Algorithm of PaPQS

The complete algorithm of the proposed PaPQS is provided in Algorithm B1.

---

**Algorithm B1** The proposed PaPQS active learning framework.

---

1: Randomly sample $N_{initial}$ IC batches and simulate their PDE solutions to build the cold-start training dataset $\mathcal{S}_{\mathrm{train}}$.
2: Train neural surrogate $\mathcal{M}_\theta$ with $\mathcal{S}_{\mathrm{train}}$ by SWAG to update surrogate parameter distribution $p(\theta)$
3: **for** $itr \in$ active learning iteration **do**
4:      Obtain $2^{itr-1} \times N_{initial}$ batches of ICs $\mathcal{S}_{\mathrm{batch}}^{(0)} = \{\boldsymbol{\psi}_1, \cdots, \boldsymbol{\psi}_{N_{\mathrm{batch}}}\}$.
5:      Obtain the samples of $\theta$ by Eq. (10) and joint samples $\{(\theta_i, \mathbf{u}(\boldsymbol{\psi}_i, t; \theta_i)) \mid \theta_i \sim p(\theta), \boldsymbol{\psi}_i \in \mathcal{S}_{\mathrm{batch}}^{(0)}, t \in [1 : N_t]\}$. Permute joint samples to get independent samples.
6:      Train neural critic function $T_\phi(\cdot)$ by maximizing Eq. (9).
7:      **for** each $\mathcal{S}_{\mathrm{batch}}$ **do**
8:        **for** $n \in$ total update steps $N_{steps}$ **do**
9:          **for** each $t \in [1 : N_t]$ **do**
10:            Predict $\mathbf{u}(\boldsymbol{\psi}_{\mathrm{batch}}, t; \theta)$ and estimate EIG by Eq. (9).
11:            **if** $t \in \{1, \lfloor N_t/2 \rfloor, N_t\}$ **then**
12:              Calculate the gradient by Eqs. (12)-(13)
13:            **end if**
14:          **end for**
15:          Average the gradients across time steps.
16:          Conduct the gradient ascent as by Eq. (5)
17:          Calculate the policy score by Eq. (7) and determine whether to retain each update.
18:        **end for**
19:      **end for**
20:      Simulate all updated batches of ICs to get the ground truth and add them to $\mathcal{S}_{\mathrm{train}}$
21:      Train neural surrogate $\mathcal{M}_\theta$ with $\mathcal{S}_{\mathrm{train}}$ by SWAG to update parameter distribution $p(\theta)$
22: **end for**

---

## C  Details of PDEs and Neural Surrogates

In this section, we describe the neural PDE solvers evaluated in our study in Sec. 4 of the main text, along with the procedures used to generate their initial conditions and corresponding hyperparameters. We note that our settings are aligned with the AL4PDE benchmark[1] [13] for consistency and fair comparison.

---

[1] https://github.com/dmusekamp/al4pde

## C.1 PDEs and their parameters

**Burgers' equation.** We consider the one-dimensional (1D) viscous Burgers' equation, a canonical nonlinear PDE used to model wave propagation and shock formation:

$$\partial_t \mathbf{u} + \mathbf{u}\,\partial_x \mathbf{u} = \frac{\nu}{\pi}\,\partial_{xx}\mathbf{u}, \tag{18}$$

where $\mathbf{u}$ denotes the velocity field, and $\nu > 0$ is the viscosity coefficient. The equation is equipped with periodic boundary conditions and initialized by a prescribed function $\mathbf{u}(x, 0)$. Following previous benchmarks [13, 23, 28], we sample PDE parameters on a logarithmic scale. Specifically, each parameter $\lambda_i$ is first uniformly sampled from $[0, 1)$ and then transformed to its domain $[a, b)$ via

$$\lambda_i = a_i \exp\left(\log\left(\frac{b_i}{a_i}\right)\lambda_i\right). \tag{19}$$

**Kuramoto-Sivashinsky (KS) equation.** The KS equation is a nonlinear fourth-order PDE that models spatiotemporal chaos in systems such as flame fronts and thin film flows. In one dimension, it takes the form:

$$\partial_t \mathbf{u} + \mathbf{u}\,\partial_x \mathbf{u} + \partial_{xx}\mathbf{u} + \nu\,\partial_{xxxx}\mathbf{u} = 0, \tag{20}$$

where $\mathbf{u}$ denotes the scalar field over space and time, and $\nu > 0$ controls the strength of the fourth-order dissipative term. The KS equation exhibits complex nonlinear dynamics and serves as a canonical testbed for studying spatiotemporal chaos and instability-driven pattern formation. Following the AL4PDE [13] configuration, the dissipation coefficient $\nu$ is treated as a varying parameter, sampled uniformly from the range $\nu \in [0.5, 4)$.

**Combined equation (CE).** The CE is a nonlinear conservation law that generalizes several classical PDEs by combining nonlinear advection, diffusion, and dispersion [43]. It is formulated by:

$$\partial_t \mathbf{u} + \partial_x\left(\alpha \mathbf{u}^2 - \beta\,\partial_x \mathbf{u} + \gamma\,\partial_{xx}\mathbf{u}\right) = 0, \tag{21}$$

where $\mathbf{u}$ denotes the scalar field, and $\alpha$, $\beta$, and $\gamma$ are scalar coefficients. By choosing different tuples $(\alpha, \beta, \gamma)$, this formulation recovers well-known equations such as the Heat equation with $(0, 1, 0)$, Burgers' equation with $(0.5, 1, 0)$, and Korteweg-de-Vries equation with $(3, 0, 1)$. We follow the standard setup adopted in AL4PDE [13] that $(\alpha, \beta, \gamma)$ are sampled uniformly from their respective ranges, with $\alpha \in [0, 3)$, $\beta \in [0, 0.4)$, and $\gamma \in [0, 1)$.

**Compressible Navier-Stokes (CNS) equation.** The CNS equations govern the evolution of a compressible viscous fluid by conserving mass, momentum, and energy [28]. It can be written in the non-conservative form as:

$$\partial_t \rho + \nabla \cdot (\rho \mathbf{v}) = 0, \tag{22}$$

$$\rho\left(\partial_t \mathbf{v} + \mathbf{v} \cdot \nabla \mathbf{v}\right) = -\nabla p + \eta \Delta \mathbf{v} + \left(\zeta + \frac{\eta}{3}\right)\nabla(\nabla \cdot \mathbf{v}), \tag{23}$$

$$\partial_t\left(\epsilon + \frac{1}{2}\rho\|\mathbf{v}\|^2\right) + \nabla \cdot \left[\left(p + \epsilon + \frac{1}{2}\rho\|\mathbf{v}\|^2\right)\mathbf{v} - \mathbf{v} \cdot \boldsymbol{\sigma}'\right] = 0. \tag{24}$$

Here, the equation has four fields for a 2D system, $\rho$ denotes the density, $\mathbf{v}$ is the velocity vector including two components, and $p$ is the pressure. The viscosity terms involve shear viscosity $\eta$ and bulk viscosity $\zeta$, and $\boldsymbol{\sigma}'$ denotes the viscous stress tensor. We follow AL4PDE [13] and sample PDE parameters $\eta$ and $\zeta$ independently from a log-uniform distribution in the range $[10^{-4}, 10^{-1})$.

## C.2 Initial conditions

Initial conditions are generated using the force density method (FDM) based JAX simulator and the IC generator from PDEBench [28]. Each initial condition is constructed as a superposition of sinusoidal waves:

$$\mathbf{u}^0(x) = \sum_{i=1}^{N_w} A_i \sin\left(\frac{2\pi k_i x}{L} + \varphi_i\right), \tag{25}$$

where $N_w$ denotes the number of wave components, $L$ is the spatial domain length, and $A_i$, $k_i$, and $\varphi_i$ represent the amplitude, wave number, and phase of the $i$-th component, respectively.

For the Burgers' equation, we set $N_w = 2$. The amplitudes $A_i$ and phases $\varphi_i$ are sampled uniformly from $[0, 1)$ and $[0, 2\pi)$, respectively, while the wave numbers $k_i$ are integers sampled from $[1, 5)$. In addition, we apply a windowing operation by setting all values outside the interval $[x_L, x_R]$ to zero with a probability of 10%. Here, $x_L$ is sampled uniformly from $[0.1, 0.45]$ and $x_R$ from $[0.55, 0.9]$. Furthermore, the sign of $\mathbf{u}^0$ is flipped randomly with a probability of 10%.

For the KS equation, we set $N_w = 10$. The amplitudes $A_i$ and phases $\varphi_i$ are sampled uniformly from $[-1, 1)$ and $[0, 2\pi)$, respectively, while the wave numbers $k_i$ are integers sampled from $[1, 10)$.

For the CE, we set $N_w = 5$. The amplitudes $A_i$ and phases $\varphi_i$ are sampled uniformly from $[-0.4, 0.4)$ and $[0, 2\pi)$, respectively, while the wave numbers $k_i$ are integers sampled from $[1, 3)$.

For the 2D-CNS equation, the initial condition of each physical field (velocity, density, and pressure) is constructed as a superposition of 2D sinusoidal modes:

$$\mathbf{u}^0 = \mathbf{v}^0(x, y), \rho^0(x, y), p^0(x, y) = \sum_{(k_x, k_y) \in \mathcal{K}} A_{k_x, k_y} \sin\left(2\pi(k_x x + k_y y) + \varphi_{k_x, k_y}\right), \qquad (26)$$

where $\mathcal{K} = [-k_{\text{tot}}, k_{\text{tot}}] \times [-k_{\text{tot}}, k_{\text{tot}}] \setminus \{(k_x, k_y) \mid k_x = 0 \text{ or } k_y = 0\}$. We set $k_{\text{tot}} = 4$. For each mode, the amplitude is set deterministically as $A_{k_x, k_y} = 1/\sqrt[4]{k_x^2 + k_y^2}$. The corresponding phases $\varphi_{k_x, k_y}$ are uniformly sampled in the range $[0, 2\pi)$. After summing over all modes, the velocity magnitude is normalized to match a sampled target Mach number from $[0.1, 1)$. Then, we enforce the positivity of the density and pressure channels. The raw density field $\rho^0(x, y)$ is rescaled to a positive field as

$$\rho^0(x, y) = \bar{\rho}\left(1 + \Delta_\rho \frac{\rho^0(x, y)}{\max_{x,y} |\rho^0(x, y)|}\right), \qquad (27)$$

where $\bar{\rho}$ is the base density sampled from $[0.1, 10)$ and $\Delta_\rho$ from $[0.013, 0.26)$. The pressure field $p^0(x, y)$ is transformed analogously using its own scale $\Delta_p$ sampled from $[0.04, 0.8)$, and offset $\bar{p} = \bar{T}\bar{\rho}$, where $\bar{T}$ is sampled from $[0.1, 10)$. Finally, we apply a windowing operation by setting all values outside the interval $[0, 1]$ to zero with a probability of 50%.

## C.3 Grids

We consider neural PDE solvers that require discretizing both the spatial and temporal domains. The domain length and grid resolution configurations for each PDE are summarized in Table C1.

Table C1: Domain lengths and resolution configurations for each PDE. *Simulation Res.* denotes the resolution settings used in the numerical simulations, while *Neural Surrogate Res.* refers to the resolution settings used during neural PDE surrogate training.

| PDE | Temporal Domain | Spatial Domain | Simulation Res. | | Neural Surrogate Res. | |
|---|---|---|---|---|---|---|
| | | | $N_t$ | $(N_x, [N_y])$ | $N_t$ | $(N_x, [N_y])$ |
| Burgers | $[0, 2]$ | $[0, 1]$ | 201 | (1024) | 41 | (256) |
| KS | $[0, 40]$ | $[0, [0.1, 100]]$ | 801 | (512) | 41 | (256) |
| CE | $[0, 4]$ | $[0, 16]$ | 501 | (64) | 51 | (64) |
| 2D-CNS | $[0, 1]$ | $[0, 1] \times [0, 1]$ | 21 | (128, 128) | 21 | (64, 64) |

## C.4 Neural surrogates

**U-Net.** We employ an enhanced U-Net architecture [5] as one of the neural PDE surrogates. This architecture improves the traditional U-Net architecture [48] by incorporating recently developed components with improved performances in computer vision tasks, including Wide ResNet-style convolutional blocks [49], group normalization [50], and spatial attention mechanisms [51]. It additionally replaces max-pooling with downsampling layers to improve multi-scale feature extraction. To further enhance its capacity for modeling complex physical dynamics, it introduces a Fourier U-Net variant, which integrates Fourier Neural Operator layers [44] into the deeper encoder and decoder blocks. This hybrid design allows the model to efficiently capture both global and local spatial dependencies via fast Fourier transforms and mode-specific weight multiplication. Compared to

standard U-Nets, this architecture leverages Fourier representations to effectively capture multi-scale PDE dynamics, enabling generalized modeling across diverse physical systems.

**Fourier Neural Operator (FNO).** We employ FNO [44] as a neural PDE surrogate due to its efficiency and widespread adoption in scientific machine learning. FNO is designed to learn mappings between infinite-dimensional function spaces, enabling efficient solutions as surrogate modeling of complex PDEs. Unlike conventional neural networks, FNO operates in the Fourier space by parameterizing the integral kernel and applying transformations via the Fast Fourier Transform (FFT), allowing efficient modeling of global spatial dependencies with quasi-linear complexity. Its architecture lifts input functions to high-dimensional spaces, processes them through Fourier layers and nonlinear activations, and projects them back to the output domain. FNO supports mesh invariance, resolution generalization, and efficient inference, making it well-suited for modeling complex PDEs across a wide range of scientific applications.

**SineNet**. We employ SineNet [7] as another neural PDE surrogate. Conventional U-Nets, while effective for multi-scale spatial processing, suffer from temporal misalignment in skip connections for latent feature evolution across time. SineNet mitigates this issue by stacking multiple U-shaped network blocks – called waves – each advancing the solution over a small temporal interval. This multi-stage design reduces misalignment and improves modeling accuracy. Each wave combines sequential and parallel multi-scale processing through disentangled block residuals, enhancing expressiveness while maintaining parameter efficiency via adaptive channel widths. Its ability to capture complex temporal dynamics and support variable time steps makes SineNet a robust choice for scientific modeling.

**Hyperparameters**. We follow the settings provided in the AL4PDE benchmark [13]. For **U-Net**, we use the GELU activation function [52] and Fourier-based conditioning [51], with a channel multiplier of $[1, 2, 2, 4]$ and 16 hidden channels, resulting in 3.38M and 9.18M parameters for 1D and 2D versions, respectively. The implementation of **FNO** also adopts the GELU activation but uses an additional input channel for conditioning. It consists of 4 layers, with a width of 64 in 1D and 32 in 2D cases, respectively. The number of Fourier modes is set to 20. The parameter counts are 0.68M (1D) and 6.56M (2D). Lastly, for **SineNet**, we apply the GELU activation and Fourier conditioning, with 32 hidden channels and four sequential U-Net "waves". The 2D version of SineNet contains approximately 5.02M parameters.

# D Further implementation details of the proposed PaPQS

## D.1 Initial condition search space

To enable gradient-based optimization, we restrict our search space to be continuous and differentiable. The search space includes all PDE parameters and continuous scalar parameters used in constructing the initial states. Discrete or non-differentiable parameters (e.g., wave numbers and spatial windowing bounds) are excluded from the optimization process. For the Burgers' equation, we include viscosity $v$, amplitudes $A$, and phase $\varphi$. For the KS equation, we include viscosity $v$, domain length $L$, amplitudes $A$, and phase $\varphi$. For the CE, we include coefficients $\alpha$, $\beta$, and $\gamma$, amplitudes $A$, and phase $\varphi$. For the 2D-CNS equation, we include shear viscosity $\eta$, bulk viscosity $\zeta$, phase $\varphi$, Mach number, base density $\bar{\rho}$ and its scale $\Delta_\rho$, as well as base pressure and its scale $\Delta_p$.

## D.2 Neural critic function

We design a neural critic function $T_\phi(\theta, \mathbf{u}(t, \psi))$ based on the Mutual Information Neural Estimation (MINE) framework [39] to estimate mutual information (equivalent to EIG as discussed in the main text) between the last-layer parameters of neural surrogates and trajectory predictions. The critic function learns to assign a score to paired samples $(\theta, \mathbf{u}(t, \psi))$ by approximating their log-density ratio. It consists of two input branches: a parameter encoder for $\theta \in \mathbb{R}^{d_\theta}$ and a trajectory encoder for $\mathbf{u}(t, \psi) \in \mathbb{R}^{\mathcal{X} \times N_c}$, where $\mathcal{X}$ is 1D or 2D spatial grid coordinates in our study and $N_c$ is the number of physical fields.

The $\theta$-branch is a multilayer perceptron (MLP) composed of two linear layers with the ReLU activation functions that projects the input into a compact latent representation. The $\mathbf{u}$-branch employs either a 1D or 2D convolutional encoder depending on the dimensionality of the input. They both apply three successive convolutional layers followed by the ReLU activations to extract

hierarchical features from frames of PDE trajectories. The resulting features are flattened and passed through a dense projection head to match the dimensionality of the $\theta$-branch output. The outputs of both branches are concatenated and passed through a fusion MLP that outputs a scalar critic score. The detailed layer information and hyperparameters are summarized in Table D1.

The entire critic neural network is trained by minimizing the following objective:

$$- \left( \mathbb{E}_{p(\theta,\mathbf{u})}[T_\phi(\theta,\mathbf{u})] - e^{-1}\mathbb{E}_{p(\theta)p(\mathbf{u})}[e^{T_\phi(\theta,\mathbf{u})}] \right). \tag{28}$$

We train this neural critic function using mini-batch stochastic optimization on a set of joint samples $(\theta, \mathbf{u})$. The full sample set is randomly split into 80% training and 20% validation subsets. We use the Adam optimizer [33] with standard settings to minimize the MINE loss. At each active learning iteration, the neural critic function is trained for a maximum of 500 epochs. To prevent from overfitting and reduce computational cost, we employ early stopping, terminating training if the validation loss fails to improve for 20 consecutive epochs.

Table D1: Overview and hyperparameter configuration of the neural critic function. We set $d_h = 128$ in our study and *ks* refers to the kernel size of the convolutional layer.

| Component | Layer Type | #Hidden Channels | Output Dim |
|---|---|---|---|
| $\theta$ Branch | Linear + ReLU | $d_\theta \to d_h$ | $d_h$ |
| | Linear + ReLU | $d_h \to d_h/2$ | $d_h/2$ |
| **u** Branch (1D) | Conv1D + ReLU | ks=3, stride=2, $N_c \to d_h/64$ | $(d_h/64, d_x/2)$ |
| | Conv1D + ReLU | ks=3, stride=2, $d_h//64 \to d_h/32$ | $(d_h/32, d_x/4)$ |
| | Conv1D + ReLU | ks=3, stride=2, $d_h//32 \to d_h/16$ | $(d_h/16, d_x/8)$ |
| **u** Branch (2D) | Conv2D + ReLU | ks=(3,3), stride=2, $N_c \to d_h/32$ | $(d_h/32, d_x/2, d_y/2)$ |
| | Conv2D + ReLU | ks=(3,3), stride=2, $d_h/32 \to d_h/32$ | $(d_h/32, d_x/4, d_y/4)$ |
| | Conv2D + ReLU | ks=(3,3), stride=2, $d_h/32 \to d_h/16$ | $(d_h/16, d_x/8, d_y/8)$ |
| **u** Flatten Layer | Linear + ReLU | flatten size $\to$ flatten size/4 | flatten size /4 |
| | Linear + ReLU | flatten size/4 $\to d_h$ | $d_h$ |
| | Linear + ReLU | $d_h \to d_h/2$ | $d_h/2$ |
| Fusion MLP | Linear + ReLU | $d_h \to d_h$ | $d_h$ |
| | Linear + ReLU | $d_h \to d_h/4$ | $d_h/4$ |
| | Linear | $d_h/4 \to 1$ | 1 |

### D.3 Hardware and Platform

All experiments are conducted on a single GPU node of a high-performance computing cluster. The node is equipped with an Intel Xeon 6248R CPU and an NVIDIA A100 GPU with 40 GB of VRAM. Simulating, training, and evaluation are performed using Jax and PyTorch frameworks on a Linux-based operating system.

## E   Additional empirical results

### E.1   Detailed results in Fig. 2 of the main text

In Tables E1- E4, we provide all the RMSE and standard deviation values at different quantiles by neural PDE surrogates for different PDE systems, which have been used to prepare for Fig. 2 in the main text. For the Burgers' equation, each method was evaluated over 10 independent runs with different random seeds, while for the other systems, results are obtained over 5 runs. Again, it is clear that applying PaPQS can consistently achieve the best or second best approximation to the ground-truth PDE solutions across nearly all tested PDE systems.

### E.2   Exemplar neural PDE solution approximations

We present several representative case studies to illustrate the effectiveness of our PaPQS active learning framework. Specifically, we compare prediction results with and without PaPQS on two

PDE systems: the KS and the 2D-CNS equations. These systems are selected due to their complex dynamics, chaotic behavior, and high nonlinearities, which pose significant challenges for neural surrogate modeling. The visual comparisons in Figs. E1- E4 highlight how PaPQS improves prediction accuracy and stability in these scenarios through gradient-guided query synthesis iterations.

Table E1: Error metrics on Burgers' equation.

| Iteration | 1 | 2 | 3 | 4 | 5 |
|---|---|---|---|---|---|
| RMSE $\times 10^{-2}$ | | | | | |
| Random | 3.684±1.203 | 3.278±2.107 | 1.607±0.485 | 1.062±0.614 | 0.552±0.133 |
| SBAL | 3.684±1.203 | 1.179±0.223 | 0.586±0.106 | 0.400±0.075 | 0.259±0.028 |
| LCMD | 3.684±1.203 | 0.808±0.053 | 0.521±0.052 | 0.394±0.043 | 0.269±0.014 |
| Core-Set | 3.684±1.203 | 1.021±0.160 | 0.659±0.100 | 0.476±0.134 | 0.292±0.015 |
| Top-K | 3.684±1.203 | 1.494±0.250 | 0.964±0.258 | 0.477±0.044 | 0.360±0.096 |
| BAIT | 3.684±1.203 | 0.903±0.138 | 0.537±0.030 | 0.392±0.035 | 0.266±0.024 |
| LHS | 3.441±1.708 | 1.930±0.300 | 1.354±0.529 | 1.057±0.539 | 0.521±0.117 |
| Random+PaPQS | 3.684±1.203 | 1.316±0.712 | 0.808±0.123 | 0.531±0.112 | 0.358±0.115 |
| SBAL+PaPQS | 3.684±1.203 | 0.908±0.342 | 0.515±0.088 | **0.334±0.042** | **0.231±0.022** |
| LCMD+PaPQS | 3.684±1.203 | **0.757±0.226** | **0.469±0.107** | 0.379±0.101 | 0.235±0.018 |
| 50% Quantile $\times 10^{-2}$ | | | | | |
| Random | 0.182±0.015 | 0.122±0.015 | 0.083±0.010 | 0.058±0.005 | 0.044±0.007 |
| SBAL | 0.182±0.015 | 0.178±0.032 | 0.105±0.011 | 0.078±0.011 | 0.054±0.006 |
| LCMD | 0.182±0.015 | 0.129±0.014 | 0.101±0.015 | 0.068±0.008 | 0.050±0.006 |
| Core-Set | 0.182±0.015 | 0.169±0.017 | 0.133±0.013 | 0.094±0.014 | 0.063±0.008 |
| Top-K | 0.182±0.015 | 0.197±0.020 | 0.176±0.024 | 0.109±0.010 | 0.078±0.012 |
| BAIT | 0.182±0.015 | 0.150±0.014 | 0.115±0.011 | 0.079±0.006 | 0.058±0.008 |
| LHS | 0.174±0.014 | **0.116±0.014** | 0.081±0.009 | 0.062±0.007 | 0.054±0.011 |
| Random+PaPQS | 0.182±0.015 | 0.119±0.010 | **0.072±0.007** | **0.054±0.008** | **0.043±0.008** |
| SBAL+PaPQS | 0.182±0.015 | 0.163±0.031 | 0.099±0.017 | 0.078±0.008 | 0.059±0.008 |
| LCMD+PaPQS | 0.182±0.015 | 0.129±0.038 | 0.106±0.019 | 0.094±0.022 | 0.052±0.007 |
| 95% Quantile $\times 10^{-2}$ | | | | | |
| Random | 1.468±0.136 | 0.834±0.125 | 0.502±0.037 | 0.343±0.014 | 0.255±0.025 |
| SBAL | 1.468±0.136 | 1.054±0.248 | 0.544±0.065 | 0.409±0.064 | 0.269±0.026 |
| LCMD | 1.468±0.136 | **0.669±0.069** | 0.526±0.064 | 0.347±0.030 | 0.259±0.032 |
| Core-Set | 1.468±0.136 | 0.865±0.123 | 0.662±0.090 | 0.503±0.113 | 0.259±0.020 |
| Top-K | 1.468±0.136 | 1.273±0.177 | 1.045±0.200 | 0.575±0.064 | 0.449±0.077 |
| BAIT | 1.468±0.136 | 0.800±0.160 | 0.532±0.045 | 0.378±0.021 | 0.274±0.026 |
| LHS | 1.390±0.142 | 0.803±0.114 | 0.474±0.038 | 0.344±0.027 | **0.246±0.024** |
| Random+PaPQS | 1.468±0.136 | 0.762±0.075 | **0.473±0.083** | **0.324±0.073** | 0.265±0.039 |
| SBAL+PaPQS | 1.468±0.136 | 0.889±0.178 | 0.503±0.095 | 0.396±0.073 | 0.277±0.045 |
| LCMD+PaPQS | 1.468±0.136 | 0.683±0.189 | 0.495±0.117 | 0.456±0.107 | 0.261±0.026 |
| 99% Quantile $\times 10^{-2}$ | | | | | |
| Random | 6.315±0.838 | 3.327±0.724 | 1.653±0.111 | 0.968±0.046 | 0.649±0.027 |
| SBAL | 6.315±0.838 | 3.169±0.945 | 1.360±0.213 | 0.987±0.239 | 0.599±0.056 |
| LCMD | 6.315±0.838 | 1.802±0.157 | 1.223±0.237 | **0.819±0.108** | 0.573±0.041 |
| Core-Set | 6.315±0.838 | 2.461±0.500 | 1.756±0.360 | 1.153±0.295 | 0.703±0.056 |
| Top-K | 6.315±0.838 | 4.456±1.685 | 3.251±1.039 | 1.347±0.129 | 1.048±0.326 |
| BAIT | 6.315±0.838 | 2.371±0.718 | 1.255±0.108 | 0.853±0.065 | 0.612±0.055 |
| LHS | 6.215±1.012 | 3.017±0.476 | 1.515±0.109 | 0.963±0.046 | 0.650±0.047 |
| Random+PaPQS | 6.315±0.838 | 2.756±1.141 | 1.573±0.387 | 0.943±0.146 | 0.631±0.125 |
| SBAL+PaPQS | 6.315±0.838 | 2.182±0.774 | 1.177±0.232 | 0.867±0.262 | 0.606±0.045 |
| LCMD+PaPQS | 6.315±0.838 | **1.779±0.376** | **1.114±0.164** | 1.024±0.241 | **0.571±0.038** |

Table E2: Error metrics on KS

| Iteration | 1 | 2 | 3 | 4 | 5 |
|---|---|---|---|---|---|
| RMSE | | | | | |
| Random | 0.452±0.026 | 0.370±0.012 | 0.312±0.013 | 0.272±0.010 | 0.229±0.010 |
| SBAL | 0.452±0.026 | 0.347±0.020 | 0.281±0.010 | 0.236±0.008 | 0.200±0.012 |
| LCMD | 0.452±0.026 | 0.370±0.009 | 0.315±0.013 | 0.266±0.019 | 0.219±0.018 |
| Core-Set | 0.452±0.026 | 0.389±0.011 | 0.335±0.013 | 0.278±0.006 | 0.235±0.020 |
| Top-K | 0.452±0.026 | 0.378±0.018 | 0.305±0.011 | 0.264±0.014 | 0.225±0.015 |
| BAIT | 0.452±0.026 | 0.368±0.017 | 0.294±0.016 | 0.240±0.009 | 0.205±0.011 |
| LHS | 0.439±0.008 | 0.369±0.024 | 0.316±0.011 | 0.270±0.009 | 0.222±0.012 |
| Random+PaPQS | 0.452±0.026 | 0.344±0.019 | 0.298±0.012 | 0.246±0.009 | 0.203±0.007 |
| SBAL+PaPQS | 0.452±0.026 | **0.313±0.017** | **0.261±0.006** | **0.221±0.008** | **0.183±0.009** |
| LCMD+PaPQS | 0.452±0.026 | 0.354±0.011 | 0.315±0.009 | 0.247±0.019 | 0.201±0.010 |
| 50% Quantile | | | | | |
| Random | 0.021±0.005 | 0.011±0.002 | 0.008±0.001 | **0.005±0.001** | **0.003±0.001** |
| SBAL | 0.021±0.005 | 0.016±0.004 | 0.013±0.003 | 0.008±0.001 | 0.006±0.001 |
| LCMD | 0.021±0.005 | 0.020±0.003 | 0.016±0.003 | 0.009±0.003 | 0.006±0.001 |
| Core-Set | 0.021±0.005 | 0.020±0.003 | 0.016±0.003 | 0.009±0.003 | 0.009±0.002 |
| Top-K | 0.021±0.005 | 0.020±0.003 | 0.018±0.002 | 0.012±0.003 | 0.010±0.002 |
| BAIT | 0.021±0.005 | 0.020±0.003 | 0.015±0.003 | 0.008±0.001 | 0.005±0.001 |
| LHS | 0.019±0.001 | 0.011±0.002 | **0.007±0.001** | **0.005±0.001** | **0.003±0.001** |
| Random+PaPQS | 0.021±0.005 | **0.010±0.001** | **0.007±0.001** | **0.005±0.001** | **0.003±0.001** |
| SBAL+PaPQS | 0.021±0.005 | 0.014±0.003 | 0.012±0.001 | 0.008±0.001 | 0.006±0.001 |
| LCMD+PaPQS | 0.021±0.005 | 0.017±0.002 | 0.014±0.002 | 0.008±0.008 | 0.006±0.001 |
| 95% Quantile | | | | | |
| Random | 0.603±0.106 | 0.363±0.020 | 0.231±0.024 | 0.143±0.011 | 0.094±0.006 |
| SBAL | 0.603±0.106 | 0.376±0.060 | 0.255±0.031 | 0.163±0.022 | 0.119±0.018 |
| LCMD | 0.603±0.106 | 0.458±0.024 | 0.344±0.024 | 0.230±0.035 | 0.140±0.223 |
| Core-Set | 0.603±0.106 | 0.501±0.025 | 0.425±0.034 | 0.295±0.021 | 0.213±0.053 |
| Top-K | 0.603±0.106 | 0.458±0.017 | 0.340±0.026 | 0.257±0.039 | 0.188±0.016 |
| BAIT | 0.603±0.106 | 0.450±0.051 | 0.269±0.043 | 0.163±0.020 | 0.100±0.012 |
| LHS | 0.572±0.020 | 0.352±0.065 | 0.238±0.027 | 0.148±0.016 | 0.091±0.006 |
| Random+PaPQS | 0.603±0.106 | 0.321±0.021 | **0.216±0.030** | **0.135±0.009** | **0.085±0.006** |
| SBAL+PaPQS | 0.603±0.106 | **0.317±0.031** | 0.221±0.011 | 0.157±0.019 | 0.105±0.009 |
| LCMD+PaPQS | 0.603±0.106 | 0.426±0.050 | 0.331±0.033 | 0.198±0.023 | 0.132±0.017 |
| 99% Quantile | | | | | |
| Random | 2.368±0.153 | 1.844±0.105 | 1.382±0.117 | 1.040±0.092 | 0.708±0.048 |
| SBAL | 2.368±0.153 | 1.655±0.137 | 1.177±0.100 | 0.844±0.103 | 0.619±0.093 |
| LCMD | 2.368±0.153 | 1.811±0.056 | 1.440±0.097 | 1.151±0.123 | 0.802±0.149 |
| Core-Set | 2.368±0.153 | 1.920±0.077 | 1.571±0.090 | 1.230±0.046 | 0.982±0.202 |
| Top-K | 2.368±0.153 | 1.860±0.126 | 1.356±0.092 | 1.138±0.086 | 0.873±0.119 |
| BAIT | 2.368±0.153 | 1.782±0.112 | 1.265±0.131 | 0.863±0.051 | 0.607±0.055 |
| LHS | 2.296±0.053 | 1.844±0.160 | 1.426±0.089 | 1.036±0.082 | 0.667±0.058 |
| Random+PaPQS | 2.368±0.153 | 1.686±0.173 | 1.337±0.104 | 0.918±0.081 | 0.607±0.040 |
| SBAL+PaPQS | 2.368±0.153 | **1.422±0.150** | **1.046±0.061** | **0.778±0.088** | **0.545±0.066** |
| LCMD+PaPQS | 2.368±0.153 | 1.714±0.097 | 1.470±0.038 | 1.011±0.142 | 0.701±0.052 |

Table E3: Error metrics on CE.

| Iteration | 1 | 2 | 3 | 4 | 5 |
|---|---|---|---|---|---|
| RMSE $\times 10^{-2}$ | | | | | |
| Random | 4.651±1.293 | 3.814±1.121 | 2.609±0.466 | 1.630±0.257 | 1.108±0.117 |
| SBAL | 4.651±1.293 | 1.597±0.083 | 0.931±0.125 | 0.496±0.087 | 0.318±0.048 |
| LCMD | 4.651±1.293 | 1.528±0.121 | 0.957±0.114 | 0.609±0.107 | 0.338±0.041 |
| Core-Set | 4.651±1.293 | 1.596±0.235 | 1.033±0.076 | 0.761±0.230 | 0.424±0.053 |
| Top-K | 4.651±1.293 | 1.678±0.099 | 0.904±0.101 | 0.529±0.103 | 0.373±0.077 |
| BAIT | 4.651±1.293 | 1.415±0.187 | 0.900±0.102 | 0.660±0.159 | 0.424±0.124 |
| LHS | 5.130±0.808 | 3.626±1.011 | 2.668±0.383 | 1.852±0.301 | 1.312±0.144 |
| Random+PaPQS | 4.651±1.293 | 3.332±1.170 | 2.113±0.278 | 1.334±0.199 | 0.916±0.088 |
| SBAL+PaPQS | 4.651±1.293 | **1.333±0.092** | **0.759±0.092** | **0.451±0.067** | **0.295±0.060** |
| LCMD+PaPQS | 4.651±1.293 | 1.485±0.110 | 0.885±0.073 | 0.536±0.117 | 0.316±0.078 |
| 50% Quantile $\times 10^{-2}$ | | | | | |
| Random | 0.238±0.025 | 0.166±0.036 | 0.125±0.021 | 0.083±0.005 | 0.065±0.004 |
| SBAL | 0.238±0.025 | 0.200±0.024 | 0.125±0.009 | **0.076±0.008** | 0.052±0.004 |
| LCMD | 0.238±0.025 | 0.171±0.007 | 0.128±0.015 | 0.083±0.008 | 0.054±0.004 |
| Core-Set | 0.238±0.025 | 0.224±0.070 | 0.168±0.020 | 0.143±0.059 | 0.083±0.009 |
| Top-K | 0.238±0.025 | 0.211±0.019 | 0.155±0.016 | 0.111±0.015 | 0.073±0.006 |
| BAIT | 0.238±0.025 | 0.186±0.018 | 0.146±0.011 | 0.108±0.011 | 0.080±0.006 |
| LHS | 0.249±0.030 | **0.145±0.022** | **0.117±0.019** | 0.085±0.011 | 0.066±0.003 |
| Random+PaPQS | 0.238±0.025 | 0.173±0.020 | 0.118±0.010 | 0.085±0.004 | 0.063±0.003 |
| SBAL+PaPQS | 0.238±0.025 | 0.185±0.029 | 0.123±0.022 | 0.077±0.009 | **0.052±0.003** |
| LCMD+PaPQS | 0.249±0.030 | 0.186±0.013 | 0.131±0.005 | 0.079±0.005 | 0.058±0.004 |
| 95% Quantile $\times 10^{-2}$ | | | | | |
| Random | 2.373±0.220 | 1.619±0.222 | 1.090±0.050 | 0.695±0.039 | 0.516±0.019 |
| SBAL | 2.373±0.220 | 1.723±0.126 | 0.980±0.070 | **0.510±0.036** | 0.313±0.014 |
| LCMD | 2.373±0.220 | **1.485±0.121** | 1.038±0.087 | 0.609±0.061 | 0.361±0.020 |
| Core-Set | 2.373±0.220 | 1.902±0.379 | 1.389±0.126 | 1.102±0.469 | 0.598±0.095 |
| Top-K | 2.373±0.220 | 1.586±0.101 | 1.236±0.099 | 0.739±0.151 | 0.489±0.048 |
| BAIT | 2.373±0.220 | 1.567±0.152 | 1.038±0.085 | 0.581±0.070 | 0.405±0.051 |
| LHS | 2.537±0.213 | 1.516±0.098 | 1.080±0.098 | 0.709±0.057 | 0.530±0.013 |
| Random+PaPQS | 2.373±0.220 | 1.676±0.214 | 1.016±0.152 | 0.717±0.132 | 0.505±0.013 |
| SBAL+PaPQS | 2.373±0.220 | 1.588±0.187 | **0.919±0.099** | 0.515±0.048 | **0.303±0.015** |
| LCMD+PaPQS | 2.373±0.220 | 1.495±0.110 | 0.992±0.073 | 0.566±0.015 | 0.371±0.010 |
| 99% Quantile $\times 10^{-2}$ | | | | | |
| Random | 10.192±1.523 | 7.260±1.226 | 4.741±0.281 | 2.893±0.227 | 1.870±0.099 |
| SBAL | 10.192±1.523 | 4.756±0.215 | 2.701±0.251 | 1.433±0.170 | 0.896±0.053 |
| LCMD | 10.192±1.523 | **4.198±1.015** | 2.787±0.291 | 1.571±0.212 | 1.036±0.066 |
| Core-Set | 10.192±1.523 | 5.056±0.827 | 3.526±0.276 | 2.638±1.068 | 1.446±0.292 |
| Top-K | 10.192±1.523 | 5.382±0.373 | 3.174±0.181 | 1.756±0.448 | 0.972±0.092 |
| BAIT | 10.192±1.523 | 4.290±0.307 | 2.896±0.141 | 1.939±0.172 | 1.301±0.104 |
| LHS | 10.785±1.740 | 6.863±0.578 | 4.778±0.272 | 3.090±0.546 | 1.874±0.056 |
| Random+PaPQS | 10.192±1.523 | 6.959±1.364 | 4.305±0.642 | 2.747±0.187 | 1.781±0.047 |
| SBAL+PaPQS | 10.192±1.523 | 4.463±0.583 | **2.485±0.182** | **1.424±0.073** | **0.866±0.049** |
| LCMD+PaPQS | 10.192±1.523 | 4.364±0.204 | 2.727±0.163 | 1.568±0.102 | 1.025±0.022 |

Table E4: Error metrics on 2D CNS.

| Iteration | 1 | 2 | 3 | 4 | 5 |
|---|---|---|---|---|---|
| **RMSE** | | | | | |
| Random | 2.662±0.339 | 2.162±0.029 | 1.856±0.106 | 1.572±0.072 | 1.362±0.065 |
| SBAL | 2.662±0.339 | 1.979±0.226 | 1.790±0.203 | 1.458±0.140 | 1.205±0.027 |
| LCMD | 2.662±0.339 | 1.991±0.293 | 1.734±0.189 | 1.356±0.081 | 1.277±0.083 |
| Core-Set | 2.662±0.339 | 2.322±0.350 | 1.731±0.168 | 1.613±0.202 | 1.343±0.186 |
| Top-K | 2.662±0.339 | 2.169±1.129 | 2.070±0.368 | 1.623±0.524 | 1.313±0.106 |
| BAIT | 2.662±0.339 | 2.167±0.164 | 1.715±0.269 | 1.426±0.209 | 1.234±0.126 |
| LHS | 2.459±0.081 | 2.134±0.148 | 1.829±0.098 | 1.514±0.059 | 1.344±0.038 |
| Random+PaPQS | 2.662±0.339 | 2.104±0.144 | 1.780±0.073 | 1.541±0.081 | 1.313±0.070 |
| SBAL+PaPQS | 2.662±0.339 | **1.899±0.149** | **1.621±0.151** | 1.408±0.136 | **1.174±0.041** |
| LCMD+PaPQS | 2.662±0.339 | 1.943±0.258 | 1.645±0.136 | **1.318±0.203** | 1.235±0.084 |
| **50% Quantile** | | | | | |
| Random | 0.506±0.119 | 0.447±0.156 | 0.356±0.111 | 0.266±0.087 | **0.209±0.034** |
| SBAL | 0.506±0.119 | 0.480±0.116 | 0.543±0.344 | 0.336±0.063 | 0.295±0.053 |
| LCMD | 0.506±0.119 | 0.574±0.361 | 0.412±0.234 | 0.317±0.065 | 0.312±0.085 |
| Core-Set | 0.506±0.119 | 0.562±0.154 | 0.411±0.085 | 0.433±0.191 | 0.408±0.120 |
| Top-K | 0.506±0.119 | 0.653±0.165 | 0.521±0.133 | 0.483±0.174 | 0.400±0.065 |
| BAIT | 0.506±0.119 | 0.637±0.336 | 0.392±0.076 | 0.335±0.069 | 0.311±0.093 |
| LHS | 0.553±0.132 | 0.503±0.068 | **0.304±0.035** | 0.264±0.066 | 0.233±0.041 |
| Random+PaPQS | 0.506±0.119 | **0.429±0.092** | 0.322±0.039 | **0.253±0.040** | 0.220±0.061 |
| SBAL+PaPQS | 0.506±0.119 | 0.461±0.115 | 0.341±0.074 | 0.265±0.046 | 0.259±0.080 |
| LCMD+PaPQS | 0.506±0.119 | 0.567±0.142 | 0.375±0.231 | 0.267±0.072 | 0.268±0.103 |
| **95% Quantile** | | | | | |
| Random | 4.421±0.630 | 3.491±0.154 | 2.828±0.314 | 2.317±0.207 | 1.927±0.170 |
| SBAL | 4.421±0.630 | 3.308±0.550 | 2.936±0.370 | 2.310±0.349 | 1.821±0.128 |
| LCMD | 4.421±0.630 | 3.263±0.561 | 2.758±0.351 | 2.025±0.177 | 2.003±0.326 |
| Core-Set | 4.421±0.630 | 4.235±0.899 | 2.952±0.375 | 2.690±0.396 | 2.189±0.437 |
| Top-K | 4.421±0.630 | 5.009±2.402 | 3.891±0.921 | 2.911±1.392 | 2.238±0.289 |
| BAIT | 4.421±0.630 | 3.700±0.263 | 2.783±0.547 | 2.238±0.404 | 1.900±0.273 |
| LHS | 4.173±0.299 | 3.283±0.240 | 2.840±0.230 | 2.250±0.102 | 1.926±0.087 |
| Random+PaPQS | 4.421±0.630 | 3.479±0.156 | 2.797±0.281 | 2.267±0.167 | 1.932±0.186 |
| SBAL+PaPQS | 4.421±0.630 | **3.216±0.449** | **2.518±0.432** | 2.150±0.403 | **1.809±0.203** |
| LCMD+PaPQS | 4.421±0.630 | 3.361±0.347 | 2.640±0.333 | **1.961±0.245** | 1.905±0.183 |
| **99% Quantile** | | | | | |
| Random | 11.378±1.863 | 9.135±0.253 | 7.754±0.507 | 6.620±0.340 | 5.735±0.320 |
| SBAL | 11.378±1.863 | 8.295±1.062 | 7.195±0.786 | 6.058±0.573 | 4.933±0.112 |
| LCMD | 11.378±1.863 | 8.196±0.926 | 7.229±0.609 | 5.569±0.362 | 5.265±0.399 |
| Core-Set | 11.378±1.863 | 9.739±1.416 | 7.263±0.707 | 6.646±0.794 | 5.404±0.722 |
| Top-K | 11.378±1.863 | 11.424±5.585 | 8.531±1.478 | 6.466±2.101 | 5.237±0.417 |
| BAIT | 11.378±1.863 | 8.948±0.487 | 7.140±1.168 | 5.923±0.922 | 5.059±0.598 |
| LHS | 10.422±0.367 | 8.800±0.769 | 7.727±0.531 | 6.374±0.198 | 5.611±0.132 |
| Random+PaPQS | 11.378±1.863 | 8.851±0.798 | 7.732±0.423 | 6.689±0.364 | 5.916±0.370 |
| SBAL+PaPQS | 11.378±1.863 | 8.275±0.722 | **6.911±0.750** | 5.985±0.879 | **4.859±0.857** |
| LCMD+PaPQS | 11.378±1.863 | **7.944±1.581** | 6.992±0.582 | **5.235±0.403** | 5.156±0.443 |

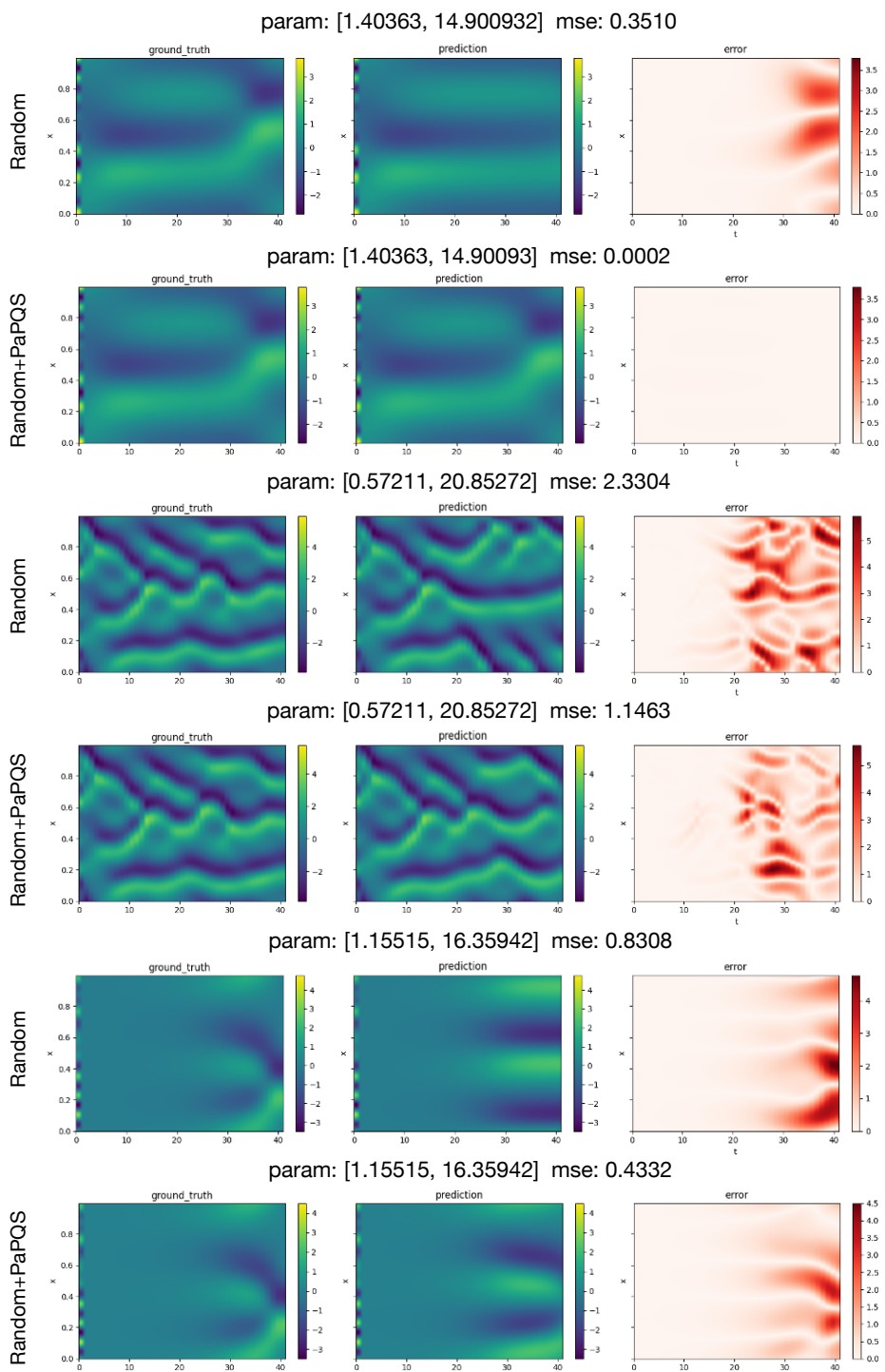

Figure E1: Neural PDE solution examples on the KS equation. The left, center, and right columns display the ground truth, model prediction, and absolute error, respectively. The x-axis represents time, while the y-axis corresponds to the spatial domain.

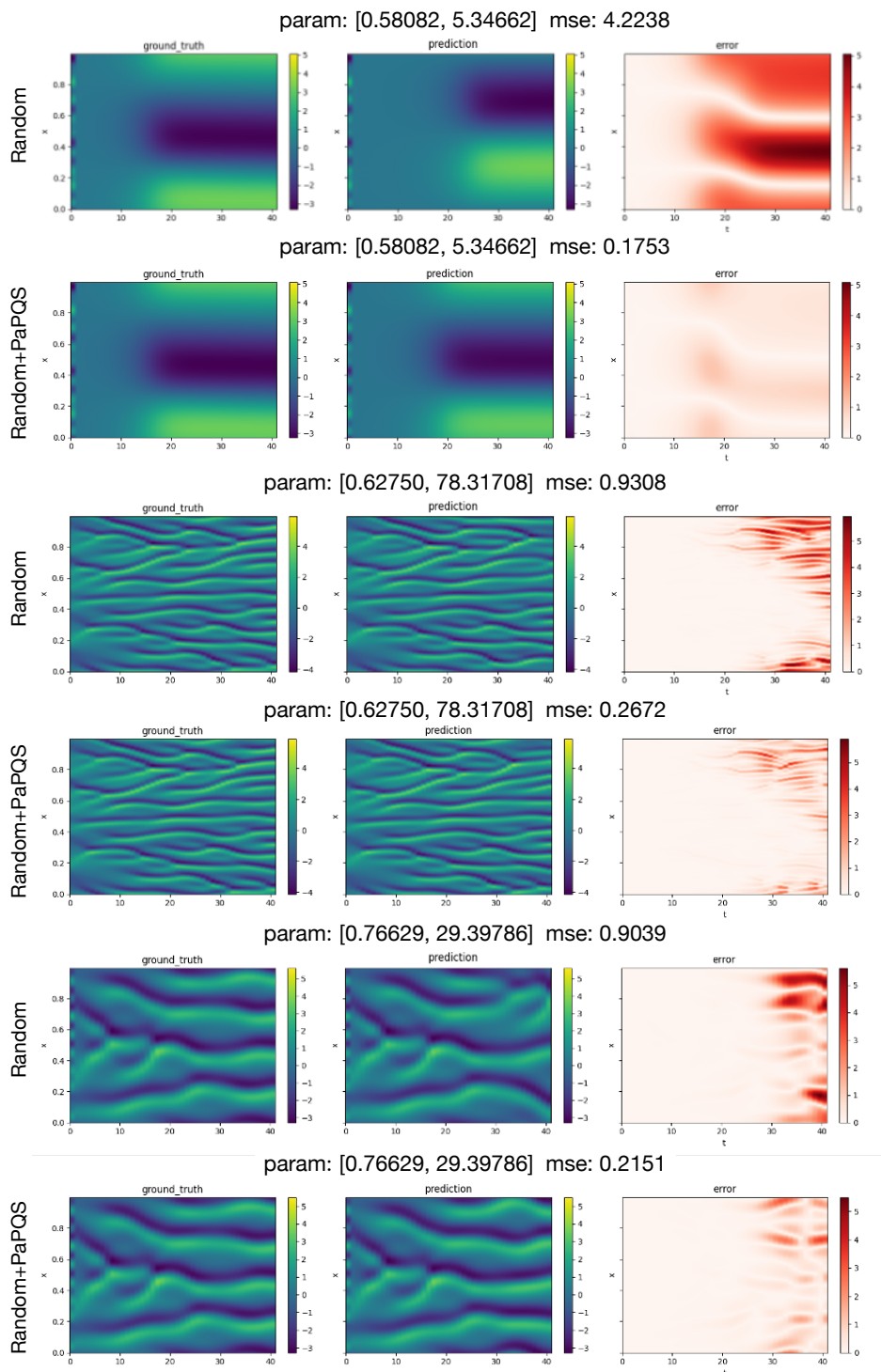

Figure E2: Neural PDE solution examples on the KS equation. The left, center, and right columns display the ground truth, model prediction, and absolute error, respectively. The x-axis represents time, while the y-axis corresponds to the spatial domain.

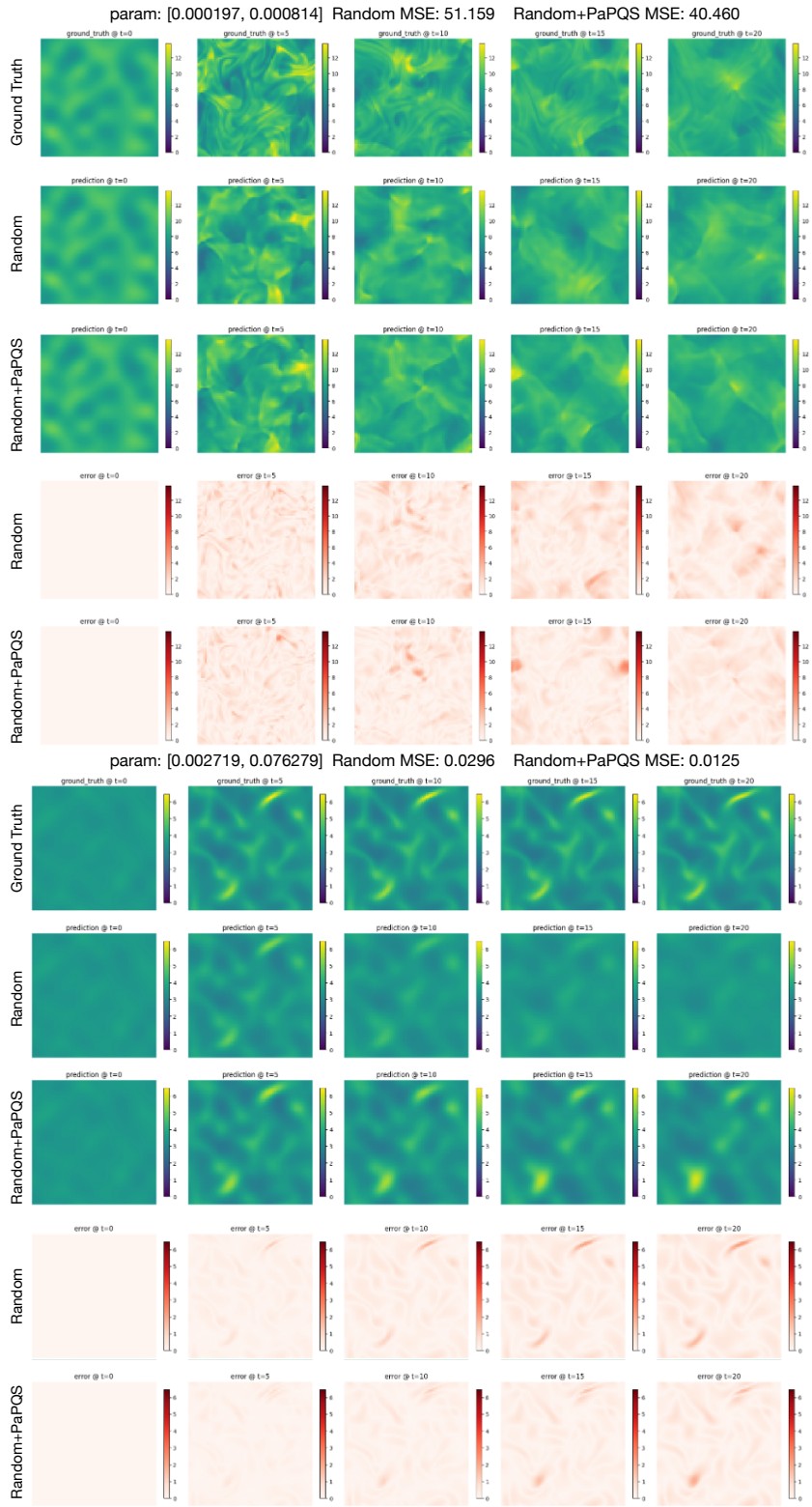

Figure E3: Neural PDE solution examples on the 2D-CNS equation. Each column displays snapshots at time steps $t = 0, 5, 10, 15, 20$. For each example, the first row shows the ground truth. The second and third rows compare neural surrogate approximations by the models trained with random sampling and PaPQS, respectively. The fourth and fifth rows compare the corresponding absolute approximation error.

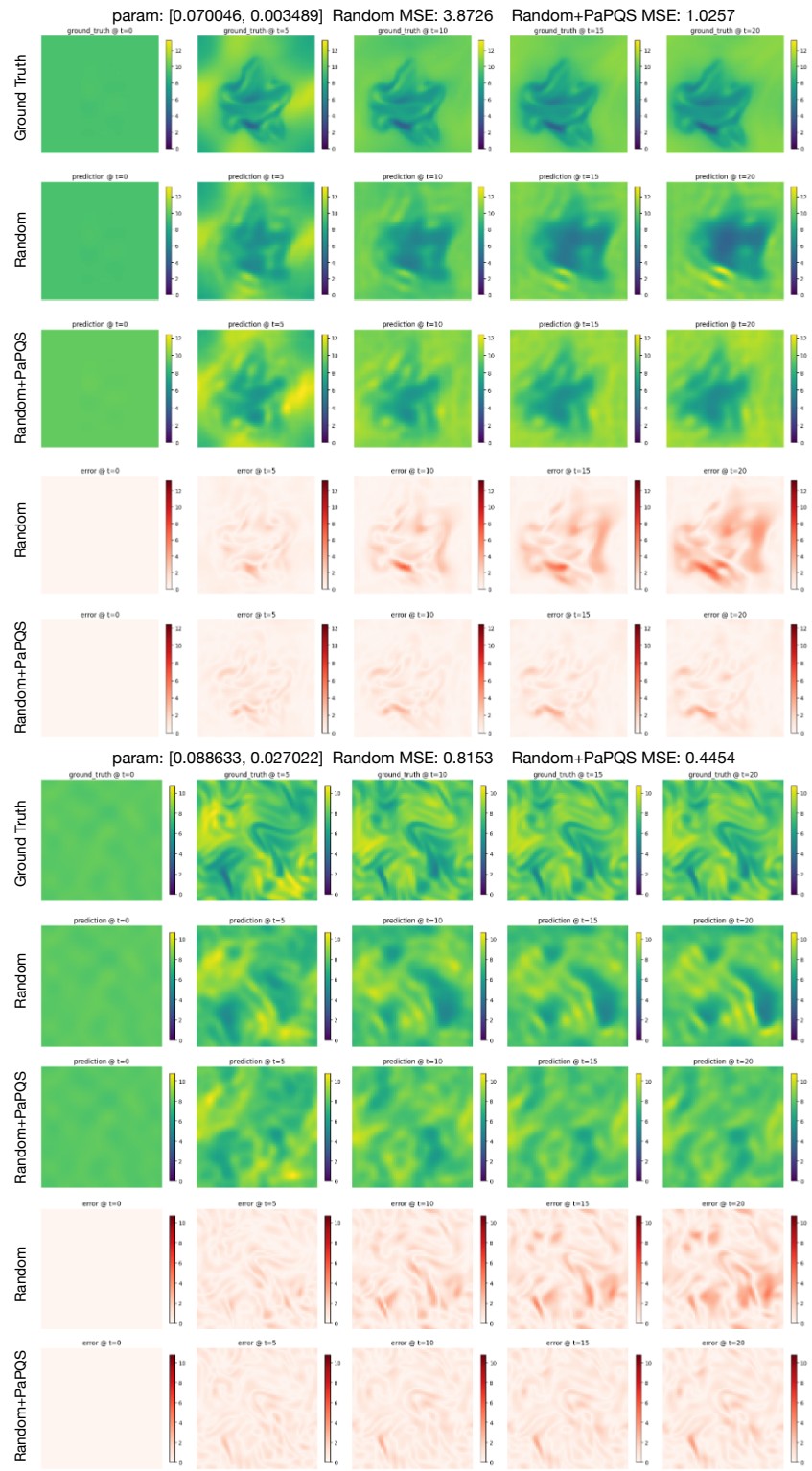

Figure E4: Neural PDE solution examples on the 2D-CNS equation. Each column displays snapshots at time steps $t = 0, 5, 10, 15, 20$. For each example, the first row shows the ground truth. The second and third rows compare neural surrogate approximations by the models trained with random sampling and PaPQS, respectively. The fourth and fifth rows compare the corresponding absolute approximation error.

