# OpenReview forum: "A Plug-and-Play Query Synthesis Active Learning Framework for Neural PDE Solvers"
_NeurIPS.cc/2025/Conference — NeurIPS 2025 poster_

### Official Review · Reviewer_kYvd · 2025-06-27

**Clarity:** 2
**Significance:** 3
**Originality:** 2
**Rating:** 3
**Confidence:** 3

**Summary:**

The work addresses the learning of an autoregressive model for time series prediction in a context where data generation is expensive. The paper investigates learning tasks on physical time series as a representative example of this problem, where a simulator generates temporal trajectories with a natural trade-off between simulation quality and speed. To improve learning efficiency, the paper proposes an active learning approach for selecting initial conditions—not randomly, but in a way that ensures the resulting trajectories most benefit the model.

Specifically, the selected initial conditions are those expected to induce the greatest change in the current model. This approach is formalized within a probabilistic Bayesian framework. A key challenge lies in identifying, in advance and without time evolution, which initial conditions are likely to contribute the most information to the existing set of trajectories. To address this, a second model, referred to as the critic, is introduced. The critic does not require autoregressive rollout and can also provide estimates when the likelihood function of the autoregressive model is inaccessible. The entire system is jointly optimized, including the parameters of the autoregressive model, the initial conditions for new trajectories, and the critic’s parameters. The method is then evaluated through a series of experiments, including several differential equations, various active learning strategies, and different model architectures.

**Questions:**

- L57: A sentence in the paper ‘By explicitly targeting the reduction of the posterior uncertainty, our framework dynamically explores high-informative regions while avoiding instability-prone area’. Can you explain how to draw this conclusion? I wonder if it’s not exactly the opposite that’s the case. Shouldn’t an unstable trajectory maximize the expected information gain as processing an unstable trajectory would result in a significantly different model?

- Are there any physics-specific components or assumptions in the method? If not, should the approach also be evaluated on non-physical time series tasks to better demonstrate its generality?

- How does the method perform on stochastic time series?

Suggestions:

- As pointed out, a theoretical justification could really strengthen the paper. A study on a linear system (linear physics and linear model) may allow for a closed-form solution, which could reveal if the proposed method provides a lower variance estimator for the model parameters.

- Another empirical test could be to visualize the optimization landscape. Most likely, the optimization landscape provided by the critic is smoother, especially in the beginning. Similarly, the spectrum of the Hessian indicates how fast gradient-based optimization finds solutions. This could be another strategy to gain some insights on the proposed method.

Minor:
L98: MSE loss, not RMSE
L212: typo ‘neural’

**Ethical Concerns:**

["NO or VERY MINOR ethics concerns only"]

**Final Justification:**

Post rebuttal: my main concern is that we are dealing with three nested nonlinear optimization problems. These are commonly approached by iterative methods that utilize the current local structure of the problem, as you noted, and I do not question that this is appropriate. My issue, however, is that due to this complicated setup, it is difficult to compare methods that solve it and conclude that one is better than another, which you claim in your paper. For me, a purely experimental evaluation is not enough given this complex setup.

A theoretical analysis, as provided in your references [2,3] (rebuttal not paper), captures one part, namely the active learning. However, there is still the iterative updating of the model and the critic. Assume linear function approximation for both of them, then you enter a regime where gradient descent is well studied and hence, where all three components are individually understood. My suggestion is to put these ideas together for your setup, where then 3 nested linear problems still form an overall linear problem. The way this multiplies out depends on the specifics of your method. If the final expression shows your method's advantage, then, combined with the experimental results, this will be a strong paper.

I wouldn't oppose if others argue for acceptance, but I'm inclined to keep my current score of a weak reject.

**Limitations:**

Yes

**Paper Formatting Concerns:**

-

**Quality:**

2

**Strengths And Weaknesses:**

I am uncertain about the novelty of the proposed approach and the use of critic networks in active learning. Critic networks are common in other areas to process information over time more effectively, such as reinforcement learning.

My main concern is that the overall setup seems overly complex, involving three nested non-convex optimization problems, each sensitive to initialization and hyperparameters. Even with extensive experimentation, I find it difficult to draw a conclusion about general applicability here.

The authors argue (lines 179–182) that small perturbations can lead to spikes in predictive variance due to the chaotic nature of the system and the autoregressive structure, thereby justifying their more complicated approach. But more generally, it is questionable if autoregressive setups or application of learning to chaotic system is a suitable task for approximate machine learning, because of that sensitivity. One could just as easily argue that the expected information gain is misled by small perturbations of a labeled initial condition that leads to a completely different trajectory and so drastically changes the model?

I encourage the authors to explore theoretical approaches to better understand and justify their method. As noted, a purely empirical evaluation seems of limited value in this context. However, when combined with a solid theoretical analysis, a more compelling case could be made.

Strengths

- Relevant topic and direction of research

- Many experiments and a detailed statistical evaluation

Weaknesses:

- Highly complex non-convex optimization

- The introduction of the critic network feels abrupt—it appears in the middle of the technical derivation without sufficient context. As mentioned, the central challenge is to estimate the usefulness of a specific initial condition without evolving the system, and this is addressed using the critic. This connection should be communicated more clearly. Both the abstract and introduction should mention that a second neural network is involved and plays a key role in the method.

- In my view, the derivation from the expected information gain (Equation 11) to the update equations (Equation 13) is difficult to follow. The text primarily describes what is being done, but the reasons behind the design choices are not sufficiently explained. Any additional intuition or explanation would be helpful here. It might also improve clarity to restructure Sections 3.1  (optimizer and loss term 2/regularization term)  and 3.2 (loss term 1/EIG and update equations). Also, devoting half a page to the Adam optimizer seems unnecessary, as it is a standard choice.

- Equation 4, in my opinion, does not provide much insight, and the mathematical notation is somewhat confusing.

- Regarding Claim 1 and Proposition 1, the statements are not mathematically precise. For example: “Assume that the dominant sources of epistemic uncertainty of the model can be captured by θ_out” is vague and would benefit from clarification. Additionally, the final step in the proof (Equation 18) should be explicitly referenced in the appendix as well. Since this step appears to be crucial, an intuitive explanation of the identity in the main text could help readers better understand the overall argument.

---

> ### Author Rebuttal · Authors · 2025-07-31
>
> We appreciate your thoughtful comments.
>
> >W1: Highly complex non-convex optimization & theoretical justification on linear system
>
> We acknowledge that our acquisition optimization is a non-convex problem, which is a common and often unavoidable characteristic for active learning with complex models and in more general optimal experimental design under uncertainty, especially with high-dimensional and structured design spaces such as PDE initial condition space. The common solution strategies typically follow gradient-based methods, which is what we adopted with our proposed gradient estimates based on EIG acquisition function. We acknowledge that such an optimization solution may not lead to global optimal solutions. Nevertheless, our optimization converges reliably in practice with standard optimizers (e.g., Adam) at a moderate computational cost. Empirically, we observe consistent improvements across benchmarks (see Tables E1–E4), indicating that the local optima found by our method are sufficiently effective.
>
> Theoretical analysis of EIG-based methods and active learning for linear systems, indicated by the reviewer, has been studied in literature, including references [2][3].  If the model is linear, the scenario simplifies to a linear projection of the output, resulting in maximizing the determinant of identity plus the Hessian of the misfit. Consequently, in this case, the focus directly shifts to reducing uncertainty with respect to the model parameter covariance matrix.
>
> However, theoretical analysis to extend to PDE neural solvers can be challenging as commented by the reviewer. Our work instead adopts a practical approach by approximating EIG using a neural critic trained with the MINE loss. This enables scalable acquisition in complex PDE settings, where theoretical guarantees are difficult to derive. We agree that extending theoretical analysis to more general nonlinear systems (e.g., constrained surrogates or kernel-based approximations) would be a valuable direction for future research, and we will discuss our limitations and this future direction in the final version.
>
> > W2: The introduction of the critic network feels abrupt.…
>
> We will clarify in the abstract that our method uses a neural critic function to estimate the EIG.
>
> Revised version:
>
> To improve the utility of candidate settings, PaPQS optimizes the Expected Information Gain (EIG) estimated by a learned neural critic for each PDE setting without evolving the PDE system, enabling model-aware exploration of the design space via backpropagation through the neural PDE solution trajectories. Meanwhile, it encourages batch diversity to prevent sampled PDE settings from overconcentrating in narrow regions of the input space.
>
> We will also supplement the relative description in the introduction in the final version.
>
> > W3: The derivation problem in Sec. 3.2. Devoting half a page to the Adam optimizer seems unnecessary.
>
> Section 3.2 follows the following organization: acquisition objective - required sampling - optimization procedure. Firstly, we introduce the acquisition function based on EIG. We then describe how we estimate EIG using the MINE-f lower bound (Eq. 11), followed by posterior sampling via SWAG (Eq. 12) to enable sampling of $\theta$, which are required for the bound computation. Finally, we provide the gradient formulation for optimizing the initial conditions $\psi$ through gradient ascent (Eq. 13), including how we backpropagate through the neural PDE solver.
>
> To improve clarity, we will better connect our design choices and strengthen the intuition for our gradient-based acquisition process. We will shorten the Adam part and instead use the space to elaborate on the motivations behind our EIG formulation and its differentiability, which we believe will significantly improve the readability and justification of our method.
>
> > W4: Eq. 4 does not provide much insight, and the mathematical notation is somewhat confusing.
>
> Our intention with this formulation was to concisely summarize the query synthesis procedure as a gradient-based optimization over a batch of initial conditions, which is guided by the informativeness scoring function $\pi(\cdot)$.
>
> To improve clarity, we will: (1) update the notation by removing the argmax over $\mathcal{G}$ to avoid the confusion with misled impression considering a search over operators. Instead, we will present it as a regular optimization procedure; (2) emphasize that the scoring function $\pi(\cdot)$ is explicitly defined in Eq. (9), where its components are detailed (We will also add the corresponding hyperlink); (3) provide a more intuitive explanation in the text to help readers better understand the role of Eq (4) as a summary of the iterative query synthesis process via backpropagation through both the surrogate model and the learned neural critic.
>
> > W5: Regarding Claim 1 and Proposition 1, the statements are not mathematically precise. Eq.18 should be explicitly referenced...
>
> We will explicitly reference Eq. 18 in the main text and clarify its connection to the MINE-f lower bound of mutual information. We will revise Proposition 1 as follows:
>
> ...Assume that uncertainty over surrogate predictions $\mathbf{u}$  is dominated by the output layer parameters $\theta_{out}$ , i.e., $p(\mathbf{u}|\theta,\psi)= p(\mathbf{u}|\theta_{out},\psi)$. Then, the $EIG(\psi)$ can be approximated by sampling only  $\theta_{out}$, i.e., .$EIG(\psi)=I(\theta;\mathbf{u}|\psi)=I(\theta_{out};\mathbf{u}|\psi)$.  A valid lower bound...
>
> We will introduce mathematical descriptions for the terms in Claim 1:
>
> prediction uncertainty: $Var_{\theta \sim p(\theta)}[\mathbb{E}[\mathbf{u} | \theta,\psi]]$
>
> > Q1: L57: A sentence in the paper ‘By explicitly...instability-prone area’. Explain how to draw this conclusion...
>
> > One could just as easily argue that the EIG is misled by small perturbations of a labeled initial condition that leads to a completely different trajectory and so drastically changes the model?
>
> We agree that in chaotic systems, small perturbations can cause large variations in model outputs, which can in turn yield high predictive variance or even numerically large EIG values. However, not all this kind of uncertainty is epistemically informative. For these instances, we need to determine whether the observed uncertainty stems from a lack of knowledge about the model parameters (epistemic) or from intrinsic instability that cannot be reduced through learning (aleatoric or chaotic).
>
> Our framework solves this problem by using EIG as the acquisition objective, which explicitly aims to reduce posterior uncertainty over model parameters. This objective steers the acquisition process toward epistemically informative and learnable regions, rather than those where uncertainty primarily arises from trajectory instability. For example, in neural PDE surrogate modeling, certain initial conditions can lead to sensitive or chaotic responses, but such regions may inflate entropy without contributing useful gradients for learning. Training on these unstable trajectories may also hurt model robustness and generalization.
>
> In a word, compared to previous variance-based acquisition methods, our EIG-based approach inherently conditions on both observations and model parameters, capturing mutual informativeness. This allows PaPQS to mitigate the risk of being misled by unstable trajectories that often reflect high variance but low epistemic gain.
>
> We will revise the related presentation to make our motivation clearer. We hope this addresses the reviewer’s concern.
>
> > Q2&3: Are there physics-specific components or assumptions? Should the approach also be evaluated on non-physical time series? How does the method perform on stochastic time series?
>
> While PaPQS does not rely on any physics-specific priors or inductive biases, our current design focuses on PDE solvers and continuous initial condition spaces. This aligns with PDE-based scientific modeling, where the design space is inherently continuous and query synthesis is preferable to standard pool-based methods. This motivates the development of PaPQS as the focus of active learning is to train neural PDE solvers that can accelerate the underlying PDE solutions with all possible parameters, boundary and initial conditions in the PDE systems under study.
>
> For the review suggested, when developing active learning with non-physical time series, typical active learning benchmarks rely on fixed, pre-collected datasets and lack a natural continuous query space, and therefore most of benchmarked active learning methods are pool-based instead of considering continuous optimization as in PaPQS. This limits the direct applicability of our method in its current form. Extending PaPQS to such domains would require rethinking the query synthesis mechanism, potentially by constructing a corresponding pool-based variant, for which our developed gradient estimates, one of our methodological contributions in PaPQS, may not be needed as the active learning can be done by either ranking or greedy search based on estimated EIG.
>
> Regarding stochastic time series, our method can still be applicable, especially for those with a surrogate model that explicitly outputs a predictive distribution. Since PaPQS estimates EIG, access to the predictive likelihood enables the use of techniques such as nested Monte Carlo or contrastive methods [1] to approximate the EIG more directly. We believe this extension is promising, especially when the surrogate is probabilistic and uncertainty-aware.
>
> [1] Foster, Adam, et al. "A unified stochastic gradient approach to designing Bayesian-optimal experiments." AISTATS. PMLR, 2020.
>
> [2] Alexanderian, et al. "On Bayesian A-and D-optimal experimental designs in infinite dimensions." (2016):671-695.
>
> [3] Attia, Ahmed, et al. "Goal-oriented optimal design of experiments for large-scale Bayesian linear inverse problems." Inverse Problems,2018:095009.

---

> > ### Comment · Reviewer_kYvd · 2025-08-06
> >
> > Thank you for responding to my review.
> >
> > As mentioned previously, my main concern is that we are dealing with three nested nonlinear optimization problems. These are commonly approached by iterative methods that utilize the current local structure of the problem, as you noted, and I do not question that this is appropriate. My issue, however, is that due to this complicated setup, it is difficult to compare methods that solve it and conclude that one is better than another, which you claim in your paper. For me, a purely experimental evaluation is not enough given this complex setup.
> >
> > A theoretical analysis, as provided in your references [2,3] (rebuttal not paper), captures one part, namely the active learning. However, there is still the iterative updating of the model and the critic. Assume linear function approximation for both of them, then you enter a regime where gradient descent is well studied and hence, where all three components are individually understood. My suggestion is to put these ideas together for your setup, where then 3 nested linear problems still form an overall linear problem. The way this multiplies out depends on the specifics of your method. If the final expression shows your method's advantage, then, combined with the experimental results, this will be a strong paper. In the current form, I'm inclined to keep my current score.

---

> ### Author Response · Authors · 2025-08-04
>
> Thank you again for your time and thoughtful feedback. I noticed that you kindly acknowledged our clarifications. If you have any remaining questions or concerns that were not fully addressed, I’d be happy to clarify them further.

---

> ### Author Response · Authors · 2025-08-08
>
> Thank you for your constructive suggestion regarding a theoretical analysis under a linear setting. Following your advice, we consider a simplified surrogate model to compare the variance and EIG:
> $$
> u_{t+1}=\psi w+\epsilon,
> $$
> where $\psi$ denotes initial conditions,  $w\sim N(0, \Sigma_w)$ denotes the model parameters, and $\epsilon \sim N(0, \sigma^2)$ is the output noise, including uncertainty caused by measurement or aleatoric noise (e.g., chaos in PDEs). Now we discuss the two acquisition functions in this case.
>
> The **EIG** could be expressed as follows:
>
> $$
> EIG(\psi)=MI(w;u|\psi)=H(u|\psi)-H(u|w,\psi)
> $$
>
> Under the linear setting, we have $u|\psi \sim N(0, \psi\Sigma_w\psi^T+\sigma^2)$, $u|w,\psi \sim N(\psi w, \sigma^2)$. Then,
>
> $$
> EIG(\psi)=\frac{1}{2}\log(2\pi e(\psi\Sigma_w\psi^T+\sigma^2)) - \frac{1}{2}(2\pi e(\sigma^2))=\frac{1}{2}\log(1+\frac{\psi\Sigma_w\psi^T}{\sigma^2})
> $$
>
> This closed form shows that EIG depends on epistemic uncertainty $\psi\Sigma_w\psi^T$ related to the model parameters while invariant to additive aleatoric noise in the sense that $\sigma^2$ only acts as a normalizing constant in the log term.
>
> The **predictive variance** could be expressed as
>
> $$
> V[u|\psi]=V_{w,\epsilon}[\psi w+\epsilon]=V_{w}[\psi w] + V_\epsilon[\epsilon]=\psi\Sigma_w\psi^T+\sigma^2.
> $$
>
> Unlike EIG, the predictive variance includes both epistemic and aleatoric contributions, so it can be inflated by noise unrelated to model uncertainty.
>
>
> This shows the case that the PDE surrogate is approximated as linear functions and the critic reduces to a closed-form expression. This connects components into linear regimes as you suggested. The advantage of our approach in this simplified setting is clear: by directly approximating EIG, we focus exclusively on epistemic uncertainty and avoid being misled by aleatoric noise, including the chaotic dynamics often present in PDE systems. Such chaotic behavior can severely inflate predictive variance without increasing actual learnability, making variance-based acquisition prone to selecting uninformative regions. In contrast, our EIG-based critic naturally filters out these noisy but uninformative areas by treating noise as a normalization term, enabling more targeted and robust exploration even in highly challenging PDE regimes.
>
>
> As for the **optimization** of EIG, $\Sigma_w > 0$  ensures that the quadratic form $\psi \Sigma_\theta \psi^\top$ is convex, and the outer function $\log(1+x)$ is strictly concave and monotone increasing. By the composition rule in convex analysis, a concave and monotone increasing outer function composed with a convex inner function yields a concave overall function. Since our acquisition maximizes EIG, this becomes a convex optimization problem (maximization of a concave function), for which gradient ascent enjoys standard convergence guarantees to the global optimum in the linear case. This property supports our optimization strategy in the linear case.
>
> In the nonlinear PDE setting, deriving the exact closed-form EIG is intractable, which motivates our use of a neural critic for approximation. Nevertheless, the linear-case analysis presented here clearly demonstrates the benefit of our approach in a tractable setting, thereby providing support for our methodological choice.
>
> We will provide these detailed derivations in the Appendix, while presenting the key conclusions in the main text. We hope this will address your concern.
>
> Thanks!

---

### Official Review · Reviewer_FANU · 2025-06-28

**Clarity:** 3
**Significance:** 3
**Originality:** 4
**Rating:** 5
**Confidence:** 5

**Summary:**

The paper presents PaPQS, a query synthesis approach to sample datasets for neural PDE solvers. The method uses the mine-f method to estimate the expected information gain and SWA-Gaussian to sample the posterior distribution of model parameters. Combined with a distance-based metric for ensuring diversity, the acquisition function can be optimized using gradient ascent. The experiments show that the method can improve the samples generated by other active learning strategies.

**Questions:**

1. For combining SBAL and PaPQS, did you use EIG also as the acquisition function of SBAL or the original ensemble uncertainty?
2. Why did you apply the critic T to individual states, and not the whole trajectory at once? Will this formulation still be able to consider dynamics in the information gain?
3. Do you optimize the total number of samples to select in the given AL iteration all at once, or do you optimize within mini-batches?
In Eq. 8, did you use the distance in the initial parameter space and PDE parameters? If so, what distance measure did you use?
4. Doesn’t the acceptance of only positive steps (Eq. 9) encourage getting stuck in local minima, which may have been prevented by the momentum of Adam otherwise?
5. Minor details: In Eqs. 13 and 14, the summation index j seems not to be used within the sum. Misspelled “neural” in line 212.  In lines 133-136, you only mention batches of ICS, but shouldn’t it be batches of ICs and PDE parameters?

**Ethical Concerns:**

["NO or VERY MINOR ethics concerns only"]

**Final Justification:**

I maintain my original rating of acceptance. The presented work is novel and can improve prior pool-based techniques. The authors responded well to the mentioned weaknesses and questions.  Other reviewers pointed out mainly issues with the clarity, which I think are, however, minor,  and can be addressed in the camera-ready version.  Additionally, I agree with the other reviewers that the improvement over the respective starting AL method is relatively small. However, the difference is pretty consistently positive across the investigated PDEs and dataset sizes, which is why it still can be considered relevant.

**Limitations:**

Only one sentence, which only mentions the performance on the 2D PDE. The modest performance on a single PDE does not show a general problem for specifically multi-field PDEs; it could also be the higher number of spatial dimensions or other reasons. For example, the strong dependence of the performance on the initialization could also be mentioned.

**Paper Formatting Concerns:**

None.

**Quality:**

3

**Strengths And Weaknesses:**

Strengths
1. A novel method that can be added on top of every neural PDE solver and is faster than estimating uncertainty using ensembles.
2. One of the rare works of AL that explores the query synthesis setting.
3. Extensive experiments with confidence intervals.

Weaknesses
1. The method needs good initialization by a different AL method. If being optimized from scratch, it does not reach the performance of other AL strategies.
2. The discussion of related work could be improved. For example, [1] and [2]  could be mentioned as applications of AL for neural PDE solvers.  In ln. 46-48: “feature-based acquisition methods emphasize input diversity but do not account for the relationship between PDE settings and model parameters” is not substantiated. For the first part, the feature-based acquisition function focuses on the coverage in feature-space and not in the inputs, and is also able to consider informativeness, for example [3]. Could you explain the second part about the relationship between PDE settings and model parameters? Don’t feature-based methods also use the gradient w.r.t. to the parameters as the feature representation [3]?  Lastly, PDERefiner (reference 11 in the paper)  is cited in ln. 34 as an example for AL, but this paper does not discuss the matter.

[1] Li, S., Yu, X., Xing, W., Kirby, R., Narayan, A., & Zhe, S. (2024, April). Multi-resolution active learning of Fourier neural operators. In International Conference on Artificial Intelligence and Statistics (pp. 2440-2448). PMLR.
[2] Kim, Y., Kim, H., Ko, G., & Lee, J. Active Learning with Selective Time-Step Acquisition for PDEs. In Forty-second International Conference on Machine Learning.
[3] Holzmüller, D., Zaverkin, V., Kästner, J., & Steinwart, I. (2023). A framework and benchmark for deep batch active learning for regression. Journal of Machine Learning Research, 24(164), 1-81.

---

> ### Author Rebuttal · Authors · 2025-07-31
>
> We appreciate your thoughtful questions and feedback.
>
> > Weakness: The discussion of related work could be improved. For example, [1] and [2] could be mentioned as applications of AL for neural PDE solvers. In ln. 46-48: “feature-based acquisition methods emphasize input diversity but do not account for the relationship between PDE settings and model parameters” is not substantiated. For the first part, the feature-based acquisition function focuses on the coverage in feature-space and not in the inputs, and is also able to consider informativeness, for example [3]. Could you explain the second part about the relationship between PDE settings and model parameters? Don’t feature-based methods also use the gradient w.r.t. to the parameters as the feature representation [3]? Lastly, PDERefiner (reference 11 in the paper) is cited in ln. 34 as an example for AL, but this paper does not discuss the matter.
>
> We truly appreciate the reviewer's constructive suggestions as well as provided references. We will cite these references and provide the corresponding discussions in our revised introduction and background sections.
>
> About the first part, in l46-48, certain feature-based acquisition methods incorporate model parameters structurally by measuring in the embedding layer or introducing gradient kernels [3]. But we meant to explain that these methods only consider the relationships in the corresponding feature space implicitly, without explicitly modeling the mutual dependence between candidate inputs and model parameters, nor targeting the reduction of posterior uncertainty over parameters. We will revise the text to clarify our intention.
>
> About the second part,  [3] indeed incorporate model gradients with respect to parameters with $\nabla_\theta f(x)$ to define a similarity kernel, which allows them to measure a kind of sensitivity of predictions to model parameters and select diverse samples via strategies like MaxDiag, MaxDet, and LCMD optimization. However, our intended relationship between PDE settings and model pamaeters targets a direct posterior uncertainty reduction over model parameters. It is achieved by EIG and could be formulated by $
> \text{EIG}(\psi) = \mathbb{E}_{p(u, \theta \mid \psi)}
> \left[
> \log \frac{p(\theta \mid {u}, \psi)}{p(\theta)}
> \right]$.
>
> In other words, they use gradients of parameters as static features for diversity, while our method enables us to select settings that are more directly aligned with the goal of improving parameter estimation.
>
> [3] Holzmüller, D., Zaverkin, V., Kästner, J., & Steinwart, I. (2023). A framework and benchmark for deep batch active learning for regression. Journal of Machine Learning Research, 24(164), 1-81.
>
> > Question 1: For combining SBAL and PaPQS, did you use EIG also as the acquisition function of SBAL or the original ensemble uncertainty?
>
> In the SBAL + PaPQS combination, we retain the original ensemble-based predictive variance as the acquisition function for SBAL. PaPQS is applied after the initial candidates are proposed by SBAL, serving as further continuous optimization that refines the candidate set by encouraging informativeness (via EIG) and diversity. This two-stage setup allows PaPQS to remain compatible with various upstream acquisition functions while improving their sample efficiency.
>
> > Question 2: Why did you apply the critic T to individual states, and not the whole trajectory at once? Will this formulation still be able to consider dynamics in the information gain?
>
> We apply the critic to individual predicted states $u_t$ instead of the entire trajectory $u_{1:T}$ primarily for two reasons: (1) practical tractability considering high resolutions along the time dimension and (2) sufficient expressiveness of dynamics given the surrogate’s design.
>
> Although we apply the critic to individual predicted states $u_t$, the formulation still retains information about the full trajectory dynamics through two mechanisms. First, our surrogate model is autoregressive, where each state $u_t$ is predicted based on the previous state  $u_{t-1}$; this recursive structure ensures that the temporal dependencies and system dynamics are implicitly encoded in each step.
>
> Second, the model input at each step includes the initial condition parameters of $u_0$, meaning the influence of the starting point propagates directly through the entire rollout and is embedded into each prediction. This propagation, carried through from the initial setting, is fully available to the neural critic. Together, these two designs ensure that the neural critic function can meaningfully reflect the informativeness over the trajectory, even though it is applied at the individual state.
>
> > Question 3: Do you optimize the total number of samples to select in the given AL iteration all at once, or do you optimize within mini-batches? In Eq. 8, did you use the distance in the initial parameter space and PDE parameters? If so, what distance measure did you use?
>
> We use mini-batches, which we set to 64 for 2DCNS equations and 128 for others.
> We compute distances in the initial condition parameter space. Take the Burgers’ equation as an example, the parameter space includes the viscosity coefficient, amplitude, wavenumber, and phase.
>  For the inherently circular phase parameter, we adopt a squared circular distance defined as $\left( \min\left( \left| \phi_1 - \phi_2 \right|, 2\pi - \left| \phi_1 - \phi_2 \right| \right) \right)^2$.  We use the Euclidean distance for the other parameters. We also apply min–max normalization to each dimension based on its known bounds to ensure consistency across different parameters.
>
> > Question 4: Doesn’t the acceptance of only positive steps (Eq. 9) encourage getting stuck in local minima, which may have been prevented by the momentum of Adam otherwise?
>
> Our update criterion in Eq. (10) is not intended to replace optimizer dynamics such as momentum, but rather to combine EIG score and diversity in a tractable way. We clarify our rationale as follows:
>
> 1. Non-differentiable diversity term: The diversity regularization $\mathcal{H}(\cdot)$ is kNN-based and non-differentiable, making it incompatible with gradient-based optimizers like Adam. To incoperate this with the EIG-based acquisition term $\mathcal{A}(\cdot)$, we adopt this filtering mechanism where only updates that may improve the joint objective (informativeness + diversity) are retained.
> 2. As shown in our experiments (the table below), incorporating diversity via this hybrid policy improves performance in data-scarce regimes, suggesting that our approach effectively balances local improvement and global coverage.
>
> | Method         | iteration 0   | iteration 1   | iteration 2   |
> | -------------- | ------------- | ------------- | ------------- |
> | with diversity | 3.684 ± 1.203 | 1.316 ± 0.712 | 0.808 ± 0.123 |
> | w/o diversity  | 3.684 ± 1.203 | 1.454 ± 0.763 | 0.920 ± 0.132 |
>
> 1. As we mentioned in line 271, the threshold $\eta$ is not fixed but dynamically chosen as the median within the current batch. It means half updates are retained at each update step. This allows us to adaptively retain high-quality updates relative to the optimization landscape, rather than applying a static cutoff that could hinder exploration.
>
> > Question 5: Minor details: In Eqs. 13 and 14, the summation index j seems not to be used within the sum. Misspelled “neural” in line 212. In lines 133-136, you only mention batches of ICS, but shouldn’t it be batches of ICs and PDE parameters?
>
> Thank you for pointing out these details. We will carefully check the typos and correct them in the final version.
>
> For the Eqs. (13) and (14), we will correct the corresponding summation by the
>
> $$\frac{e^{-1}}{N_{\text{batch}}}\sum_{j=1}^{N_{\text{batch}}} e^{T_\phi(\theta_i, \mathbf{u}_j(t, \psi_j; \theta_i))}$$
> and
>
> $$ \frac{e^{-1}}{N_{\text{batch}}} \sum_{j=1}^{N_{\text{batch}}} \nabla_{\phi} e^{T_\phi(\theta_i, \mathbf{u}_j(t, \psi_j; \theta_i))}$$, respectively.
>
> In lines 133-136, it should be a batch of initial states $\mathbf{u}^0$ and PDE parameters. We will clarify the components of the initial conditions (ICs) in the Background section of the final version.

---

> > ### Comment · Reviewer_FANU · 2025-08-01
> >
> > Thank you for your clarification. I will maintain my already positive score. Do you have an explanation for why the optimization does not reach the same performance irrespective of the starting point?

---

> > > ### Author Response · Authors · 2025-08-03
> > >
> > > Thanks for your thoughtful question. Our optimization does not always reach the same performance irrespective of the starting point, mainly due to the non-convex nature of the acquisition landscape and the limited gradient steps taken during optimization. Since we aim for high efficiency, especially in high-dimensional PDE settings, we currently use a small number of gradient ascent steps to reduce computational cost. This makes the optimization sensitive to initialization. In Fig. 4(c) and lines 338-345, we also discussed the influence of the number of update steps and its potential as a tradeoff between performance and computational resources.
> > >
> > > Additionally, due to the inherent randomness in selecting training data (e.g., through random initialization or sampling noise), variability in performance can arise. For instance, on the CE dataset, we observed that the RMSE gap between the best and worst seed under the SBAL framework can exceed 25%.
> > >
> > > In future work, we plan to investigate better trade-offs between computational efficiency and convergence robustness, including (1) designing a more expressive and efficient critic network that allows us to conduct large-scale gradient-guided search, and (2) exploring adaptive strategies for determining the number of gradient steps based on the convergence situation.

---

### Official Review · Reviewer_vMpD · 2025-06-28

**Clarity:** 3
**Significance:** 2
**Originality:** 3
**Rating:** 4
**Confidence:** 3

**Summary:**

The paper introduces PaPQS, an active learning framework for neural partial differential equation solvers. Unlike traditional pool-based methods that are restricted to specific surrogate types, PaPQS synthesizes new initial conditions directly from the continuous input space. This approach enables finer control over informativeness and diversity in the query batch, helping avoid redundancy and instability. The framework is plug-and-play and generalizes well across multiple surrogate architectures, as demonstrated by comprehensive experiments.

**Questions:**

- Can you compare PaPQS with non-active learning pipelines that share the same objective goal with your paper. For example, the methods that aim for faster convergence, such as those leveraging pretraining or transfer learning?

**Ethical Concerns:**

["NO or VERY MINOR ethics concerns only"]

**Limitations:**

The authors bring up the limitations in current design choices, such as the use of fixed step sizes and the absence of physics-informed or temporally adaptive strategies.

**Paper Formatting Concerns:**

There is no major formatting issue, but there is a typo on line 237: "commonlhy".

**Quality:**

2

**Strengths And Weaknesses:**

Strengths
- The plug-and-play design is easy to integrate and broadly applicable across different neural PDE surrogates, including U-Net, SineNet, and FNO.
- The author provides systematic and well-controlled experiments that cover key aspects of the active learning pipeline, including efficiency, architecture generalizability, and data reusability.

Weaknesses
- The method requires access to the ground-truth PDE solver, making it inapplicable in settings where only precomputed or real-world datasets are available without simulation capability.
- The performance gains are modest in several cases; for example, RMSE improvements over SBAL are minimal on Burgers, 2D-CNS, and CE equations. This experiment results cannot prove the efficacy of the proposed PaPQS method.

---

> ### Author Rebuttal · Authors · 2025-07-31
>
> We appreciate your thoughtful questions and feedback.
>
> > The method requires access to the ground-truth PDE solver, making it inapplicable in settings where only precomputed or real-world datasets are available without simulation capability.
>
> Our method indeed assumes access to a ground-truth PDE solver, which allows evaluating arbitrary initial conditions in a continuous input space. This is crucial for our query synthesis framework, which actively optimizes over this space to find informative PDE settings. In contrast, precomputed or real-world datasets typically provide only a *fixed, finite* set of inputs, making it infeasible to apply our optimization-based approach directly. Therefore, our current method is not directly suited for such settings without constructing the corresponding pool-based active learning strategies as we explained in our responses to Reviewer kYvd. We would like to emphasize here that our motivation of developing PaPQS is to train neural PDE solvers that can accelerate the underlying PDE solutions with all possible parameters, boundary, and initial conditions in the PDE systems under study. This setup aligns with the nature of many PDE-based scientific modeling tasks, where the input space is inherently continuous and query synthesis is more appropriate than commonly adopted pool-based active learning.
>
> > The performance gains are modest in several cases; for example, RMSE improvements over SBAL are minimal on Burgers, 2D-CNS, and CE equations. This experiment results cannot prove the efficacy of the proposed PaPQS method.
>
> While we acknowledge that the RMSE improvements of PaPQS over SBAL are relatively modest on specific tasks such as Burgers', CE, and 2D-CNS PDE systems, we would like to emphasize the following:
>
> 1. Even in PDE benchmarks where SBAL already performs strongly, PaPQS consistently achieves additional improvement (+12.48% on Burgers, +10.26% on CE, +6.35% on KS, and +3.90% on 2D-CNS). This trend demonstrates the robustness and generality of PaPQS across different model architectures and domains. PaPQS, as a plug-in refinement module that operates on top of existing acquisition methods, has its value as a general enhancement strategy.
> 2. Current implementation uses a lightweight neural critic (a shallow CNN) and a limited number of gradient descent steps without sophisticated architectural design or algorithmic tuning. Despite this simplicity, PaPQS still delivers consistent performance gains. This suggests that the effectiveness of PaPQS stems from the generality of its framework, rather than from task-specific heuristics or finely tuned components. We believe this generality provides a strong foundation. Future work can further boost performance by exploring more expressive and efficient critic networks (e.g., Fourier Neural Operator), adaptive gradient schemes, or solver-aware inductive biases without altering the overall methodology.
>
> > Can you compare PaPQS with non-active learning pipelines that share the same objective goal with your paper. For example, the methods that aim for faster convergence, such as those leveraging pretraining or transfer learning?
>
> We agree that PaPQS shares a common high-level goal with methods such as pretraining and transfer learning—that is, to accelerate convergence and reduce the sample complexity required for model training.
>
> However, PaPQS approaches this challenge from a different aspect, focusing on data acquisition rather than model initialization. Specifically, PaPQS actively synthesizes informative PDE initial conditions in a continuous design space. This enables informative data acquisition during training, which complements rather than replaces techniques like pretraining.
>
> In contrast, pretraining or transfer learning methods typically rely on external data sources or related tasks. Their effectiveness depends on the similarity between pretraining and target domains and the quality of parameters finetuning.
>
> Moreover, PaPQS can be naturally combined with pretraining-based surrogates. For instance, one may start with a pretrained model and apply PaPQS to select the most valuable data points for fine-tuning, thereby improving sample efficiency and reducing redundant evaluations. This makes PaPQS particularly attractive in scenarios where full data acquisition is expensive, and selective fine-tuning on high-value samples is desirable. Exploring such hybrid pipelines is an exciting direction for future work.

---

### Official Review · Reviewer_ZBUc · 2025-06-30

**Clarity:** 2
**Significance:** 3
**Originality:** 2
**Rating:** 4
**Confidence:** 3

**Summary:**

The paper presents a plug and play approach towards improving neural network based models which act as PDE solvers. These models rely on training data which can be collected based on the approach in the paper PaPQS. It has two components which consider expected information gain (EIG) and sample diversity together to provide data to train neural PDE surrogates. PsPQS can be used in conjunction with other active learning strategy. Empirical results are presented to support the claims.

**Questions:**

1. Fig 2. is not possible to decipher, it is clumsy and the color scheme is bad. I am not able to decode which method is doing how much better than the other. Can you make it better and more readable?
2. Where are the tables (numerical results)? It is mentioned about Table E1, E2? I don't see them in the main paper, the hyper-links don't work. Without numerical evidence, it is hard to justify the viability of the method.
3. Fig 3(a) I don't see too much benefit over random.
4. Fig 4(b) Importance of the diversity component is not prominent but it has been claimed otherwise. Can authors clarify and provide more evidence?
5. Fig 4(a) The main EIG component also doesn't seem to be independent on it's own. Can you provide more evidence?
6. Fig 2. Is there a reason to only use PaPQS with certain AL method and not others? Where is the curve for PaPQS alone?

**Ethical Concerns:**

["NO or VERY MINOR ethics concerns only"]

**Final Justification:**

The Authors rebuttal answered my questions hence I have upgraded the score

**Limitations:**

Authors do not include any limitations of their work or any negative societal impact. I think the time complexity of the proposed approach should be discussed as it is expected to be a bottleneck when making an informed choice about the training data needed.

**Paper Formatting Concerns:**

No major formatting issue.

**Quality:**

3

**Strengths And Weaknesses:**

Strength:
1. The idea presented in the paper is very clean and makes a lot of sense intuitively, the authors do a good job in providing intuition for the idea.
2. The experimental section answers a lot of interesting questions each of which have their own merit.
3. Relevant literature has been covered.

Weakness:
1. Empirical evidence is weak to support that PaPQS has noticeable benefits.
2. The presentation is hard to follow. I think a lot of things are mentioned in a non-standard manner. Please see questions section for more elaborate details.

---

> ### Author Rebuttal · Authors · 2025-07-31
>
> We appreciate your thoughtful questions and feedback.
>
> > Weakness: Empirical evidence is weak to support that PaPQS has noticeable benefits.
>
> We understand the reviewer’s concern regarding the empirical benefits of PaPQS. We respectfully note that the performance gains of PaPQS may appear visually modest in Figure 2 due to the varying error scales across the five active learning iterations. However, based on the numerical results reported in Tables E1–E4 in Appendix E, PaPQS consistently improves performance across all tasks and acquisition strategies. For example, Random+PaPQS achieves an average RMSE reduction of **48.68%** on the Burgers’ equation task, while SBAL+PaPQS and LCMD+PaPQS achieve average improvements of 15.60% and 8.18% on top of state-of-the-art active learning strategies, respectively. Across the remaining three tasks, the performance improvement is also consistent, with an average of 6.99% on KS, 12.27% on CE, and 3.79% on 2D-CNS.
>
> We also emphasize that PaPQS is designed as a plug-in module that complements rather than replacing existing acquisition strategies. The fact that PaPQS consistently improves random sampling, uncertainty-based (e.g., SBAL), and diversity-based (e.g., LCMD) active learning methods across diverse PDE systems highlights its broad applicability. These results also demonstrate the feasibility of gradient-based search over continuous initial condition spaces in PDE problems, which can inform future active learning designs that more effectively exploit the structure of such continuous domains.
>
> Last but not least, to the best of our knowledge, PaPQS is the first query synthesis method for active learning of PDE neural surrogates considering continuous design space. To address the challenges in developing efficient solution algorithms, we estimate the acquisition function based on expected information gain (EIG) through the neural critic function with a 3-layer vanilla convolutional neural network and MINE loss, which is computationally scalable but consistently outperforms state-of-the-art active learning methods on our task. We believe that further enhancing the neural critic through more expressive architectures or refined training objectives could further boost the performance of PaPQS. We leave this as a promising direction for future work.
>
> > Question 1: Fig 2. is not possible to decipher, it is clumsy and the color scheme is bad. I am not able to decode which method is doing how much better than the other.
>
> We understand the concern that the current figure may appear visually dense due to the number of methods and overlapping confidence intervals. Given the format limitations of the rebuttal period, we are unable to include a revised version. However, we have taken the following steps to enhance clarity in the final version:
>
> 1. We will display only one representative method from each category of active learning approaches (Random, LCMD, SBAL, and BAIT) as baselines. The performance of all other methods is comprehensively reported in Tables E1–E4 in Appendix E. We will also carefully select a color scheme with improved perceptual separability.
> 2. To have a clearer comparison between methods equipped with and without our PaPQS strategy, we will use distinct line styles (solid vs. dashed) and marker types (circles vs. diamonds) to differentiate their performance more effectively.
> 3. We will change the confidence intervals from 95% to 90%. This will narrow the shaded regions and make the differences between methods clearer, especially in current densely populated plots. Importantly, this change does not affect the mean RMSE trends or alter any of the conclusions.
>
> > Question 2: Where are the tables (numerical results)? It is mentioned about Table E1, E2?
>
> The appendix containing Tables E1–E4 was uploaded separately as part of the supplementary materials (within the ZIP file). This is also the reason why the hyperlinks in the main paper are currently inactive. In the final version, we will integrate the appendix into the main paper and ensure that all internal references and hyperlinks work as intended.
>
> > Question 3:  Fig 3(a) I don't see too much benefit over random.
> >
>
> We agree that the improvements in Figure 3(a) may visually appear limited. However, the numerical results (RMSE reported in the table below) behind the figure do demonstrate that PaPQS achieves consistent improvements across all model architectures, with average improvements of **8.20%** for FNO, **5.54%** for SineNet, and **2.97%** for U-Net. The relatively smaller gain for U-Net may be attributed to its capability on the CNS task, as noted in a prior study of active learning on neural PDE solvers [1]. Thus, these results confirm that PaPQS provides consistent and significant improvements and  even when used with diverse model architectures.
>
> | Methods | FNO | FNO+PaPQS | SineNet | SineNet+PaPQS | U-Net | U-Net+PaPQS |
> | --- | --- | --- | --- | --- | --- | --- |
> | Iteration 0 | 3.900 | 3.900 | 2.800 | 2.800 | 2.660 | 2.660 |
> | Iteration 1 | 3.281 | 3.156 | 2.300 | 2.200 | 2.160 | 2.100 |
> | Iteration 2 | 3.046 | 2.786 | 1.860 | 1.760 | 1.856 | 1.780 |
> | Iteration 3 | 2.708 | 2.376 | 1.579 | 1.470 | 1.572 | 1.540 |
>
>
> > Question 4: Fig 4(b) Importance of the diversity component is not prominent but it has been claimed otherwise.
>
> We understand that the effect of diversity regularization may appear subtle at a glance in Figure 4(b), especially in latter iterations. However, the numerical results show that incorporating the diversity term reduces RMSE during the earlier stages, where active learning query selection is the most critical. For instance, the RMSE at 512 samples decreases from 1.454 (without diversity) to 1.316 (with diversity), and at 1024 samples from 0.920 to 0.808, corresponding to reductions of approximately **9.5%** and **12.2%**, respectively. These early-stage gains are critical in low-resource regimes where each query must be highly informative, and they support our claim regarding the importance of the diversity term in improving sample efficiency. Furthermore, the distribution of selected samples (bottom panel of Fig. 4b) shows that diversity regularization promotes broader coverage across the viscosity domain, avoiding overconcentration in a tiny high-uncertainty region.
>
> > Question 5: Fig 4(a) The main EIG component also doesn't seem to be independent on it's own.
>
> We have done experiments for the ablation analysis of the EIG component. The table below presents the RMSE results under different variants:
>
> - **Random**: Uniform random sampling from the initial condition space, with no acquisition mechanism.
> - **PaPQS w/o EIG**: Optimization is driven solely by diversity regularization (Eq. 8); updates are accepted based on entropy increase, without estimating EIG.
> - **PaPQS w EIG only**: The EIG component is retained and used to optimize initial conditions. Diversity term is removed.
> - **PaPQS**: Our full method.
>
> | Method | Iteration 0 | Iteration 1 | Iteration 2 | Iteration 3 | Iteration 4 |
> | --- | --- | --- | --- | --- | --- |
> | Random | 3.684 ± 1.203 | 3.278 ± 2.107 | 1.607 ± 0.485 | 1.062 ± 0.614 | 0.552 ± 0.133 |
> | PaPQS w/o EIG | 3.684 ± 1.203 | 3.139 ± 2.003 | 1.612 ± 0.369 | 1.103 ± 0.494 | 0.548 ± 0.144 |
> | PaPQS with only EIG  | 3.684 ± 1.203 | 1.454 ± 0.763 | 0.920 ± 0.132 | 0.528 ± 0.092 | 0.368 ± 0.076 |
> | PaPQS | 3.684 ± 1.203 | 1.316± 0.712 | 0.808± 0.123 | 0.531 ± 0.112 | 0.358 ± 0.115 |
>
> As shown in the table, PaPQS with only EIG significantly improves over Random and PaPQS w/o EIG variants, confirming that the EIG component alone plays an essential role in improving sample efficiency, even without other strategies.
>
> > Question 6: Fig 2. Is there a reason to only use PaPQS with certain AL method and not others? Where is the curve for PaPQS alone?
>
> We clarify that the baseline *Random* indicates that we train our model with data points randomly sampled from the initial condition distribution. *Random+PaPQS* means we apply the PaPQS and start the gradient-based optimization from these purely random samples. Thus, *Random+PaPQS* represents the performance of PaPQS alone, without the involvement of other active learning strategies.
>
> In Fig. 2, we selected representative methods from distinct categories to demonstrate the compatibility and effectiveness of PaPQS in diverse active learning settings. **Random+PaPQS** shows how PaPQS performs starting from the random initial points. **SBAL+PaPQS** shows how PaPQS performs starting from the uncertainty-based sampling. **LCMD+PaPQS** shows how PaPQS performs starting from feature-based sampling. These two categories of active learning strategies have been widely adopted in recent active learning literature. Moreover, SBAL and LCMD are both the best baselines in their respective categories. Due to space constraints and computational cost, we focused on these representative baselines rather than all possible combinations.
>
> > Limitation: I think the time complexity of the proposed approach should be discussed as it is expected to be a bottleneck when making an informed choice about the training data needed.
>
> We acknowledge that PaPQS introduces additional computational overhead. In Fig. 3(d), we reported the running time of the simulation and each component of the proposed PaPQS. These results indicate that the additional computations introduced by PaPQS are negligible, adding less than **1/20** of the total runtime compared to the simulation cost of the PDE solver. Besides, Fig. 3(c) supports that our method achieves significantly lower RMSE within the same time budget, supporting its practical efficiency.
>
> [1] Musekamp, Daniel, et al. "Active learning for neural PDE solvers." arXiv preprint arXiv:2408.01536 (2024).

---

> > ### Comment · Reviewer_ZBUc · 2025-08-06
> >
> > Dear Authors,
> >
> > Thank you for the rebuttal. I am happy with the explanation provided but it is very important that the presentation of the paper drastically change (via the above promised updates) for the paper to be well received. I understand that I will not be able to verify this, but I am upgrading my score in good faith that the authors will keep their promise.
> >
> > Thank you

---

> > > ### Author Response · Authors · 2025-08-06
> > >
> > > Thank you for your constructive feedback during the review period and the upgraded score. We will implement the promised updates to improve clarity and presentation in the final version.

---

> ### Comment · Area_Chair_jGHf · 2025-08-05
>
> Dear Reviewer,
>
> As the author-reviewer discussion period is drawing to a close, we kindly ask that you respond to the authors' rebuttal.
>
> Thank you for your work.
>
> Best regards,
> Your AC

---

### Decision · Program_Chairs · 2025-09-17

**Decision:**

Accept (poster)

**Comment:**

This paper proposes PaPQS, a plug-and-play query synthesis active learning framework for neural PDE solvers. The method is novel in that it synthesizes informative PDE initial conditions directly in the continuous design space, guided by a neural critic estimating expected information gain (EIG) and a diversity regularizer. The idea is original and timely, given the growing interest in reducing simulation costs for training neural PDE surrogates.

The reviewers highlight several strengths: PaPQS is general and can be applied to multiple surrogate architectures and acquisition strategies, it consistently improves over baselines, and it introduces query synthesis for PDE settings, which has not been systematically explored before. The experiments are extensive and include ablation studies.

Concerns were raised regarding clarity of presentation, modest performance gains over strong baselines, and the complexity of the nested optimization setup. One reviewer requested a stronger theoretical justification, noting the difficulty of drawing conclusions from purely empirical evidence. The authors addressed these points in rebuttal, committing to improved presentation and providing a useful linear-case analysis clarifying why EIG avoids noise inflation seen in variance-based methods.

While some improvements are modest, the gains are consistent across PDE systems and architectures, and the method is complementary to existing strategies rather than a replacement. The reliance on a ground-truth PDE solver does limit applicability, but the setup remains relevant for scientific machine learning contexts where simulation is available.

On balance, the novelty of the approach, the broad applicability, and the consistent improvements outweigh the weaknesses. I recommend acceptance.